# Minimax Theory for High-dimensional Gaussian Mixtures with Sparse Mean Separation

**Martin Azizyan**
Machine Learning Department
Carnegie Mellon University
mazizyan@cs.cmu.edu

**Aarti Singh**
Machine Learning Department
Carnegie Mellon University
aarti@cs.cmu.edu

**Larry Wasserman**
Department of Statistics
Carnegie Mellon University
larry@stat.cmu.edu

## Abstract

While several papers have investigated computationally and statistically efficient methods for learning Gaussian mixtures, precise minimax bounds for their statistical performance as well as fundamental limits in high-dimensional settings are not well-understood. In this paper, we provide precise information theoretic bounds on the clustering accuracy and sample complexity of learning a mixture of two isotropic Gaussians in high dimensions under small mean separation. If there is a sparse subset of relevant dimensions that determine the mean separation, then the sample complexity only depends on the number of relevant dimensions and mean separation, and can be achieved by a simple computationally efficient procedure. Our results provide the first step of a theoretical basis for recent methods that combine feature selection and clustering.

## 1   Introduction

Gaussian mixture models provide a simple framework for several machine learning problems including clustering, density estimation and classification. Mixtures are especially appealing in high dimensional problems. Perhaps the most common use of Gaussian mixtures is for clustering. Of course, the statistical (and computational) behavior of these methods can degrade in high dimensions. Inspired by the success of variable selection methods in regression, several authors have considered variable selection for clustering. However, there appears to be no theoretical results justifying the advantage of variable selection in high dimensional setting.

To see why some sort of variable selection might be useful, consider clustering $n$ subjects using a vector of $d$ genes for each subject. Typically $d$ is much larger than $n$ which suggests that statistical clustering methods will perform poorly. However, it may be the case that there are only a small number of relevant genes in which case we might expect better behavior by focusing on this small set of relevant genes.

The purpose of this paper is to provide precise bounds on clustering error with mixtures of Gaussians. We consider both the general case where all features are relevant, and the special case where only a subset of features are relevant. Mathematically, we model an irrelevant feature by requiring the mean of that feature to be the same across clusters, so that the feature does not serve to differentiate the groups. Throughout this paper, we use the probability of misclustering an observation, relative to the optimal clustering if we had known the true distribution, as our loss function. This is akin to using excess risk in classification.

This paper makes the following contributions:

- We provide information theoretic bounds on the sample complexity of learning a mixture of two isotropic Gaussians with equal weight in the small mean separation setting that precisely captures the dimension dependence, and matches known sample complexity requirements for some existing algorithms. This also debunks the myth that there is a gap between

statistical and computational complexity of learning mixture of two isotropic Gaussians for small mean separation. Our bounds require non-standard arguments since our loss function does not satisfy the triangle inequality.

- We consider the high-dimensional setting where only a subset of relevant dimensions determine the mean separation between mixture components and show that learning is substantially easier as the sample complexity only depends on the sparse set of relevant dimensions. This provides some theoretical basis for feature selection approaches to clustering.

- We show that a simple computationally feasible procedure nearly achieves the information theoretic sample complexity even in high-dimensional sparse mean separation settings.

**Related Work.** There is a long and continuing history of research on mixtures of Gaussians. A complete review is not feasible but we mention some highlights of the work most related to ours.

Perhaps the most popular method for estimating a mixture distribution is maximum likelihood. Unfortunately, maximizing the likelihood is NP-Hard. This has led to a stream of work on alternative methods for estimating mixtures. These new algorithms use pairwise distances, spectral methods or the method of moments.

Pairwise methods are developed in Dasgupta (1999), Schulman and Dasgupta (2000) and Arora and Kannan (2001). These methods require the mean separation to increase with dimension. The first one requires the separation to be $\sqrt{d}$ while the latter two improve it to $d^{1/4}$. To avoid this problem, Vempala and Wang (2004) introduced the idea of using spectral methods for estimating mixtures of spherical Gaussians which makes mean separation independent of dimension. The assumption that the components are spherical was removed in Brubaker and Vempala (2008). Their method only requires the components to be separated by a hyperplane and runs in polynomial time, but requires $n = \Omega(d^4 \log d)$ samples. Other spectral methods include Kannan et al. (2005), Achlioptas and McSherry (2005) and Hsu and Kakade (2013). The latter uses clever spectral decompositions together with the method of moments to derive an effective algorithm.

Kalai et al. (2012) use the method of moments to get estimates without requiring separation between components of the mixture components. A similar approach is given in Belkin and Sinha (2010). Chaudhuri et al. (2009) give a modified $k$-means algorithm for estimating a mixture of two Gaussians. For the large mean separation setting $\mu > 1$, Chaudhuri et al. (2009) show that $n = \tilde{\Omega}(d/\mu^2)$ samples are needed. They also provide an information theoretic bound on the necessary sample complexity of any algorithm which matches the sample complexity of their method (up to log factors) in $d$ and $\mu$. If the mean separation is small $\mu < 1$, they show that $n = \tilde{\Omega}(d/\mu^4)$ samples are sufficient. Our results for the small mean separation setting give a matching necessary condition. Assuming the separation between the component means is *not* too sparse, Chaudhuri and Rao (2008) provide an algorithm for learning the mixture that has polynomial computational and sample complexity.

Most of these papers are concerned with computational efficiency and do not give precise, statistical minimax upper and lower bounds. None of them deal with the case we are interested in, namely, a high dimensional mixture with sparse mean separation.

We should also point out that the results in different papers are not necessarily comparable since different authors use different loss functions. In this paper we use the probability of misclassifying a future observation, relative to how the correct distribution clusters the observation, as our loss function. This should not be confused with the probability of attributing a new observation to the wrong component of the mixture. The latter loss does not typically tend to zero as the sample size increases. Our loss is similar to the excess risk used in classification where we compare the misclassification rate of a classifier to the misclassification rate of the Bayes optimal classifier.

Finally, we remind the reader that our motivation for studying sparsely separated mixtures is that this provides a model for variable selection in clustering problems. There are some relevant recent papers on this problem in the high-dimensional setting. Pan and Shen (2007) use penalized mixture models to do variable selection and clustering simultaneously. Witten and Tibshirani (2010) develop a penalized version of $k$-means clustering. Related methods include Raftery and Dean (2006); Sun et al. (2012) and Guo et al. (2010). The applied bioinformatics literature also contains a huge number of heuristic methods for this problem. None of these papers provide minimax bounds for the clustering error or provide theoretical evidence of the benefit of using variable selection in unsupervised problems such as clustering.

## 2 Problem Setup

In this paper, we consider the simple setting of learning a mixture of two isotropic Gaussians with equal mixing weights,[1] given $n$ data points $X_1, \ldots, X_n \in \mathbb{R}^d$ drawn i.i.d. from a $d$-dimensional mixture density function

$$p_\theta(x) = \frac{1}{2} f(x; \mu_1, \sigma^2 I) + \frac{1}{2} f(x; \mu_2, \sigma^2 I),$$

where $f(\cdot; \mu, \Sigma)$ is the density of $\mathcal{N}(\mu, \Sigma)$, $\sigma > 0$ is a fixed constant, and $\theta := (\mu_1, \mu_2) \in \Theta$. We consider two classes $\Theta$ of parameters:

$$\Theta_\lambda = \{(\mu_1, \mu_2) : \|\mu_1 - \mu_2\| \geq \lambda\}$$

$$\Theta_{\lambda,s} = \{(\mu_1, \mu_2) : \|\mu_1 - \mu_2\| \geq \lambda, \ \|\mu_1 - \mu_2\|_0 \leq s\} \subseteq \Theta_\lambda.$$

The first class defines mixtures where the components have a mean separation of at least $\lambda > 0$. The second class defines mixtures with mean separation $\lambda > 0$ along a sparse set of $s \in \{1, \ldots, d\}$ dimensions. Also, let $P_\theta$ denote the probability measure corresponding to $p_\theta$.

For a mixture with parameter $\theta$, the Bayes optimal classification, that is, assignment of a point $x \in \mathbb{R}^d$ to the correct mixture component, is given by the function

$$F_\theta(x) = \operatorname*{argmax}_{i \in \{1,2\}} f(x; \mu_i, \sigma^2 I).$$

Given any other candidate assignment function $F : \mathbb{R}^d \to \{1, 2\}$, we define the loss incurred by $F$ as

$$L_\theta(F) = \min_\pi P_\theta(\{x : F_\theta(x) \neq \pi(F(x))\})$$

where the minimum is over all permutations $\pi : \{1, 2\} \to \{1, 2\}$. This is the probability of misclustering relative to an oracle that uses the true distribution to do optimal clustering.

We denote by $\widehat{F}_n$ any assignment function learned from the data $X_1, \ldots, X_n$, also referred to as estimator. The goal of this paper is to quantify how the minimax expected loss (worst case expected loss for the best estimator)

$$R_n \equiv \inf_{\widehat{F}_n} \sup_{\theta \in \Theta} \mathbb{E}_\theta L_\theta(\widehat{F}_n)$$

scales with number of samples $n$, the dimension of the feature space $d$, the number of relevant dimensions $s$, and the signal-to-noise ratio defined as the ratio of mean separation to standard deviation $\lambda/\sigma$. We will also demonstrate a specific estimator that achieves the minimax scaling.

For the purposes of this paper, we say that feature $j$ is irrelevant if $\mu_1(j) = \mu_2(j)$. Otherwise we say that feature $j$ is relevant.

## 3 Minimax Bounds

### 3.1 Small mean separation setting without sparsity

We begin without assuming any sparsity, that is, all features are relevant. In this case, comparing the projections of the data to the projection of the sample mean onto the first principal component suffices to achieve both minimax optimal sample complexity and clustering loss.

**Theorem 1** (Upper bound). *Define*

$$\widehat{F}_n(x) = \begin{cases} 1 & \text{if } x^T v_1(\widehat{\Sigma}_n) \geq \widehat{\mu}_n^T v_1(\widehat{\Sigma}_n) \\ 2 & \text{otherwise.} \end{cases}$$

*where $\widehat{\mu}_n = n^{-1} \sum_{i=1}^n X_i$ is the sample mean, $\widehat{\Sigma}_n = n^{-1} \sum_{i=1}^n (X_i - \widehat{\mu}_n)(X_i - \widehat{\mu}_n)^T$ is the sample covariance and $v_1(\widehat{\Sigma}_n)$ denotes the eigenvector corresponding to the largest eigenvalue of $\widehat{\Sigma}_n$. If $n \geq \max(68, 4d)$, then*

$$\sup_{\theta \in \Theta_\lambda} \mathbb{E}_\theta L_\theta(\widehat{F}) \leq 600 \max\left(\frac{4\sigma^2}{\lambda^2}, 1\right) \sqrt{\frac{d \log(nd)}{n}}.$$

*Furthermore, if $\frac{\lambda}{\sigma} \geq 2 \max(80, 14\sqrt{5d})$, then*

$$\sup_{\theta \in \Theta_\lambda} \mathbb{E}_\theta L_\theta(\widehat{F}) \leq 17 \exp\left(-\frac{n}{32}\right) + 9 \exp\left(-\frac{\lambda^2}{80\sigma^2}\right).$$

We note that the estimator in Theorem 1 (and that in Theorem 3) does not use knowledge of $\sigma^2$.

**Theorem 2** (Lower bound). *Assume that $d \geq 9$ and $\frac{\lambda}{\sigma} \leq 0.2$. Then*

$$\inf_{\widehat{F}_n} \sup_{\theta \in \Theta_\lambda} \mathbb{E}_\theta L_\theta(\widehat{F}_n) \geq \frac{1}{500} \min\left\{ \frac{\sqrt{\log 2}}{3} \frac{\sigma^2}{\lambda^2} \sqrt{\frac{d-1}{n}}, \frac{1}{4} \right\}.$$

We believe that some of the constants (including lower bound on $d$ and exact upper bound on $\lambda/\sigma$) can be tightened, but the results demonstrate matching scaling behavior of clustering error with $d$, $n$ and $\lambda/\sigma$. Thus, we see (ignoring constants and log terms) that

$$R_n \approx \frac{\sigma^2}{\lambda^2} \sqrt{\frac{d}{n}}, \quad \text{or equivalently} \quad n \approx \frac{d}{\lambda^4/\sigma^4} \text{ for a constant target value of } R_n.$$

The result is quite intuitive: the dependence on dimension $d$ is as expected. Also we see that the rate depends in a precise way on the signal-to-noise ratio $\lambda/\sigma$. In particular, the results imply that we need $d \leq n$.

In modern high-dimensional datasets, we often have $d > n$ i.e. large number of features and not enough samples. However, inference is usually tractable since not all features are relevant to the learning task at hand. This sparsity of relevant feature set has been successfully exploited in supervised learning problems such as regression and classification. We show next that the same is true for clustering under the Gaussian mixture model.

### 3.2 Sparse and small mean separation setting

Now we consider the case where there are $s < d$ relevant features. Let $S$ denote the set of relevant features. We begin by constructing an estimator $\widehat{S}_n$ of $S$ as follows. Define

$$\widehat{\tau}_n = \frac{1 + \alpha}{1 - \alpha} \min_{i \in \{1,\ldots,d\}} \widehat{\Sigma}_n(i,i), \text{ where}$$

where

$$\alpha = \sqrt{\frac{6 \log(nd)}{n}} + \frac{2 \log(nd)}{n}.$$

Now let

$$\widehat{S}_n = \{i \in \{1,\ldots,d\} : \widehat{\Sigma}_n(i,i) > \widehat{\tau}_n\}.$$

Now we use the same method as before, but using only the features in $\widehat{S}_n$ identified as relevant.

**Theorem 3** (Upper bound). *Define*

$$\widehat{F}_n(x) = \begin{cases} 1 & \text{if } x_{\widehat{S}_n}^T v_1(\widehat{\Sigma}_{\widehat{S}_n}) \geq \widehat{\mu}_{\widehat{S}_n}^T v_1(\widehat{\Sigma}_{\widehat{S}_n}) \\ 2 & \text{otherwise} \end{cases}$$

*where $x_{\widehat{S}_n}$ are the coordinates of $x$ restricted to $\widehat{S}_n$, and $\widehat{\mu}_{\widehat{S}_n}$ and $\widehat{\Sigma}_{\widehat{S}_n}$ are the sample mean and covariance of the data restricted to $\widehat{S}_n$. If $n \geq \max(68, 4s)$, $d \geq 2$ and $\alpha \leq \frac{1}{4}$, then*

$$\sup_{\theta \in \Theta_{\lambda,s}} \mathbb{E}_\theta L_\theta(\widehat{F}) \leq 603 \max\left(\frac{16\sigma^2}{\lambda^2}, 1\right) \sqrt{\frac{s \log(ns)}{n}} + 220 \frac{\sigma\sqrt{s}}{\lambda} \left(\frac{\log(nd)}{n}\right)^{\frac{1}{4}}.$$

Next we find the lower bound.

**Theorem 4** (Lower bound). *Assume that $\frac{\lambda}{\sigma} \leq 0.2$, $d \geq 17$, and that $5 \leq s \leq \frac{d+3}{4}$. Then*

$$\inf_{\widehat{F}_n} \sup_{\theta \in \Theta_{\lambda,s}} \mathbb{E}_\theta L_\theta(\widehat{F}_n) \geq \frac{1}{600} \min\left\{ \sqrt{\frac{8}{45}} \frac{\sigma^2}{\lambda^2} \sqrt{\frac{s-1}{n} \log\left(\frac{d-1}{s-1}\right)}, \frac{1}{2} \right\}.$$

We remark again that the constants in our bounds can be tightened, but the results suggest that

$$\frac{\sigma}{\lambda}\left(\frac{s^2 \log d}{n}\right)^{1/4} \succ R_n \succ \frac{\sigma^2}{\lambda^2}\sqrt{\frac{s \log d}{n}},$$

$$\text{or} \quad n = \Omega\left(\frac{s^2 \log d}{\lambda^4/\sigma^4}\right) \quad \text{for a constant target value of } R_n.$$

In this case, we have a gap between the upper and lower bounds for the clustering loss. Also, the sample complexity can possibly be improved to scale as $s$ (instead of $s^2$) using a different method. However, notice that the dimension only enters logarithmically. If the number of relevant dimensions is small then we can expect good rates. This provides some justification for feature selection. We conjecture that the lower bound is tight and that the gap could be closed by using a sparse principal component method as in Vu and Lei (2012) to find the relevant features. However, that method is combinatorial and so far there is no known computationally efficient method for implementing it with similar guarantees.

We note that the upper bound is achieved by a two-stage method that first finds the relevant dimensions and then estimates the clusters. This is in contrast to the methods described in the introduction which do clustering and variable selection simultaneously. This raises an interesting question: is it always possible to achieve the minimax rate with a two-stage procedure or are there cases where a simultaneous method outperforms a two-stage procedure? Indeed, it is possible that in the case of general covariance matrices (non-spherical) two-stage methods might fail. We hope to address this question in future work.

## 4 Proofs of the Lower Bounds

The lower bounds for estimation problems rely on a standard reduction from expected error to hypothesis testing that assumes the loss function is a semi-distance, which the clustering loss isn't. However, a local triangle inequality-type bound can be shown (Proposition 2). This weaker condition can then be used to lower-bound the expected loss, as stated in Proposition 1 (which follows easily from Fano's inequality).

The proof techniques of the sparse and non-sparse lower bounds are almost identical. The main difference is that in the non-sparse case, we use the Varshamov–Gilbert bound (Lemma 1) to construct a set of sufficiently dissimilar hypotheses, whereas in the sparse case we use an analogous result for sparse hypercubes (Lemma 2). See the supplementary material for complete proofs of all results.

In this section and the next, $\phi$ and $\Phi$ denote the univariate standard normal PDF and CDF.

**Lemma 1** (Varshamov–Gilbert bound). *Let $\Omega = \{0,1\}^m$ for $m \geq 8$. There exists a subset $\{\omega_0, ..., \omega_M\} \subseteq \Omega$ such that $\omega_0 = (0, ..., 0)$, $\rho(\omega_i, \omega_j) \geq \frac{m}{8}$ for all $0 \leq i < j \leq M$, and $M \geq 2^{m/8}$, where $\rho$ denotes the Hamming distance between two vectors (Tsybakov (2009)).*

**Lemma 2.** *Let $\Omega = \{\omega \in \{0,1\}^m : \|\omega\|_0 = s\}$ for integers $m > s \geq 1$ such that $s \leq m/4$. There exist $\omega_0, ..., \omega_M \in \Omega$ such that $\rho(\omega_i, \omega_j) > s/2$ for all $0 \leq i < j \leq M$, and $M \geq \left(\frac{m}{s}\right)^{s/5} - 1$ (Massart (2007), Lemma 4.10).*

**Proposition 1.** *Let $\theta_0, ..., \theta_M \in \Theta_\lambda$ (or $\Theta_{\lambda,s}$), $M \geq 2$, $0 < \alpha < 1/8$, and $\gamma > 0$. If for all $1 \leq i \leq M$, $\mathrm{KL}(P_{\theta_i}, P_{\theta_0}) \leq \frac{\alpha \log M}{n}$, and if $L_{\theta_i}(\widehat{F}) < \gamma$ implies $L_{\theta_j}(\widehat{F}) \geq \gamma$ for all $0 \leq i \neq j \leq M$ and clusterings $\widehat{F}$, then $\inf_{\widehat{F}_n} \max_{i \in [0..M]} \mathbb{E}_{\theta_i} L_{\theta_i}(\widehat{F}_n) \geq 0.07\gamma$.*

**Proposition 2.** *For any $\theta, \theta' \in \Theta_\lambda$, and any clustering $\widehat{F}$, let $\tau = L_\theta(\widehat{F}) + \sqrt{\mathrm{KL}(P_\theta, P_{\theta'})/2}$. If $L_\theta(F_{\theta'}) + \tau \leq 1/2$, then $L_\theta(F_{\theta'}) - \tau \leq L_{\theta'}(\widehat{F}) \leq L_\theta(F_{\theta'}) + \tau$.*

We will also need the following two results. Let $\theta = (\mu_0 - \mu/2, \mu_0 + \mu/2)$ and $\theta' = (\mu_0 - \mu'/2, \mu_0 + \mu'/2)$ for $\mu_0, \mu, \mu' \in \mathbb{R}^d$ such that $\|\mu\| = \|\mu'\|$, and let $\cos\beta = \frac{|\mu^T \mu'|}{\|\mu\|^2}$.

**Proposition 3.** *Let $g(x) = \phi(x)(\phi(x) - x\Phi(-x))$. Then $2g\left(\frac{\|\mu\|}{2\sigma}\right)\sin\beta\cos\beta \leq L_\theta(F_{\theta'}) \leq \frac{\tan\beta}{\pi}$.*

**Proposition 4.** *Let $\xi = \frac{\|\mu\|}{2\sigma}$. Then $\mathrm{KL}(P_\theta, P_{\theta'}) \leq \xi^4(1 - \cos\beta)$.*

*Proof of Theorem 2.* Let $\xi = \frac{\lambda}{2\sigma}$, and define $\epsilon = \min\left\{\frac{\sqrt{\log 2}}{3}\frac{\sigma^2}{\lambda}\frac{1}{\sqrt{n}}, \frac{\lambda}{4\sqrt{d-1}}\right\}$. Define $\lambda_0^2 = \lambda^2 - (d-1)\epsilon^2$. Let $\Omega = \{0,1\}^{d-1}$. For $\omega = (\omega(1), ..., \omega(d-1)) \in \Omega$, let $\mu_\omega = \lambda_0 e_d + \sum_{i=1}^{d-1}(2\omega(i)-1)\epsilon e_i$ (where $\{e_i\}_{i=1}^d$ is the standard basis for $\mathbb{R}^d$). Let $\theta_\omega = \left(-\frac{\mu_\omega}{2}, \frac{\mu_\omega}{2}\right) \in \Theta_\lambda$.

By Proposition 4, $\mathrm{KL}(P_{\theta_\omega}, P_{\theta_\nu}) \leq \xi^4(1 - \cos\beta_{\omega,\nu})$ where $\cos\beta_{\omega,\nu} = 1 - \frac{2\rho(\omega,\nu)\epsilon^2}{\lambda^2}$, $\omega, \nu \in \Omega$, and $\rho$ is the Hamming distance, so $\mathrm{KL}(P_{\theta_\omega}, P_{\theta_\nu}) \leq \xi^4 \frac{2(d-1)\epsilon^2}{\lambda^2}$. By Proposition 3, since $\cos\beta_{\omega,\nu} \geq \frac{1}{2}$,

$$L_{\theta_\omega}(F_{\theta_\nu}) \leq \frac{1}{\pi}\tan\beta_{\omega,\nu} \leq \frac{1}{\pi}\frac{\sqrt{1+\cos\beta_{\omega,\nu}}}{\cos\beta_{\omega,\nu}}\sqrt{1-\cos\beta_{\omega,\nu}} \leq \frac{4}{\pi}\frac{\sqrt{d-1}\epsilon}{\lambda}, \text{ and}$$

$$L_{\theta_\omega}(F_{\theta_\nu}) \geq 2g(\xi)\sin\beta_{\omega,\nu}\cos\beta_{\omega,\nu} \geq g(\xi)\sqrt{1+\cos\beta_{\omega,\nu}}\sqrt{1-\cos\beta_{\omega,\nu}} \geq \sqrt{2}g(\xi)\frac{\sqrt{\rho(\omega,\nu)}\epsilon}{\lambda}$$

where $g(x) = \phi(x)(\phi(x) - x\Phi(-x))$. By Lemma 1, there exist $\omega_0, ..., \omega_M \in \Omega$ such that $M \geq 2^{(d-1)/8}$ and $\rho(\omega_i, \omega_j) \geq \frac{d-1}{8}$ for all $0 \leq i < j \leq M$. For simplicity of notation, let $\theta_i = \theta_{\omega_i}$ for all $i \in [0..M]$. Then, for $i \neq j \in [0..M]$,

$$\mathrm{KL}(P_{\theta_i}, P_{\theta_j}) \leq \xi^4\frac{2(d-1)\epsilon^2}{\lambda^2}, \quad L_{\theta_i}(F_{\theta_j}) \leq \frac{4}{\pi}\frac{\sqrt{d-1}\epsilon}{\lambda} \quad \text{and} \quad L_{\theta_i}(F_{\theta_j}) \geq \frac{1}{2}g(\xi)\frac{\sqrt{d-1}\epsilon}{\lambda}.$$

Define $\gamma = \frac{1}{4}(g(\xi) - 2\xi^2)\frac{\sqrt{d-1}\epsilon}{\lambda}$. Then for any $i \neq j \in [0..M]$, and any $\widehat{F}$ such that $L_{\theta_i}(\widehat{F}) < \gamma$,

$$L_{\theta_i}(F_{\theta_j}) + L_{\theta_i}(\widehat{F}) + \sqrt{\frac{\mathrm{KL}(P_{\theta_i}, P_{\theta_j})}{2}} < \left(\frac{4}{\pi} + \frac{1}{4}(g(\xi) - 2\xi^2) + \xi^2\right)\frac{\sqrt{d-1}\epsilon}{\lambda} \leq \frac{1}{2}$$

because, for $\xi \leq 0.1$, by definition of $\epsilon$,

$$\left(\frac{4}{\pi} + \frac{1}{4}(g(\xi) - 2\xi^2) + \xi^2\right)\frac{\sqrt{d-1}\epsilon}{\lambda} \leq 2\frac{\sqrt{d-1}\epsilon}{\lambda} \leq \frac{1}{2}.$$

So, by Proposition 2, $L_{\theta_j}(\widehat{F}) \geq \gamma$. Also, $\mathrm{KL}(P_{\theta_i}, P_{\theta_0}) \leq (d-1)\xi^4\frac{2\epsilon^2}{\lambda^2} \leq \frac{\log M}{9n}$ for all $1 \leq i \leq M$, because, by definition of $\epsilon$, $\xi^4\frac{2\epsilon^2}{\lambda^2} \leq \frac{\log 2}{72n}$. So by Proposition 1 and the fact that $\xi \leq 0.1$,

$$\inf_{\widehat{F}_n}\max_{i \in [0..M]}\mathbb{E}_{\theta_i}L_{\theta_i}(\widehat{F}_n) \geq 0.07\gamma \geq \frac{1}{500}\min\left\{\frac{\sqrt{\log 2}}{3}\frac{\sigma^2}{\lambda^2}\sqrt{\frac{d-1}{n}}, \frac{1}{4}\right\}$$

and to complete the proof we use $\sup_{\theta\in\Theta_\lambda}\mathbb{E}_\theta L_\theta(\widehat{F}_n) \geq \max_{i\in[0..M]}\mathbb{E}_{\theta_i}L_{\theta_i}(\widehat{F}_n)$ for any $\widehat{F}_n$. $\quad\square$

*Proof of Theorem 4.* For simplicity, we state this construction for $\Theta_{\lambda,s+1}$, assuming $4 \leq s \leq \frac{d-1}{4}$. Let $\xi = \frac{\lambda}{2\sigma}$, and define $\epsilon = \min\left\{\sqrt{\frac{8}{45}}\frac{\sigma^2}{\lambda}\sqrt{\frac{1}{n}\log\left(\frac{d-1}{s}\right)}, \frac{1}{2}\frac{\lambda}{\sqrt{s}}\right\}$. Define $\lambda_0^2 = \lambda^2 - s\epsilon^2$. Let $\Omega = \{\omega \in \{0,1\}^{d-1} : \|\omega\|_0 = s\}$. For $\omega = (\omega(1), ..., \omega(d-1)) \in \Omega$, let $\mu_\omega = \lambda_0 e_d + \sum_{i=1}^{d-1}\omega(i)\epsilon e_i$ (where $\{e_i\}_{i=1}^d$ is the standard basis for $\mathbb{R}^d$). Let $\theta_\omega = \left(-\frac{\mu_\omega}{2}, \frac{\mu_\omega}{2}\right) \in \Theta_{\lambda,s}$. By Lemma 2, there exist $\omega_0, ..., \omega_M \in \Omega$ such that $M \geq \left(\frac{d-1}{s}\right)^{s/5} - 1$ and $\rho(\omega_i, \omega_j) \geq \frac{s}{2}$ for all $0 \leq i < j \leq M$. The remainder of the proof is analogous to that of Theorem 2 with $\gamma = \frac{1}{4}(g(\xi) - \sqrt{2}\xi^2)\frac{\sqrt{s}\epsilon}{\lambda}$. $\quad\square$

## 5   Proofs of the Upper Bounds

Propositions 5 and 6 below bound the error in estimating the mean and principal direction, and can be obtained using standard concentration bounds and a variant of the Davis–Kahan theorem. Proposition 7 relates these errors to the clustering loss. For the sparse case, Propositions 8 and 9 bound the added error induced by the support estimation procedure. See supplementary material for proof details.

**Proposition 5.** *Let $\theta = (\mu_0 - \mu, \mu_0 + \mu)$ for some $\mu_0, \mu \in \mathbb{R}^d$ and $X_1, ..., X_n \overset{i.i.d.}{\sim} P_\theta$. For any $\delta > 0$, we have $\|\mu_0 - \widehat{\mu}_n\| \geq \sigma\sqrt{\frac{2\max(d, 8\log\frac{1}{\delta})}{n}} + \|\mu\|\sqrt{\frac{2\log\frac{1}{\delta}}{n}}$ with probability at least $1 - 3\delta$.*

**Proposition 6.** *Let* $\theta = (\mu_0 - \mu, \mu_0 + \mu)$ *for some* $\mu_0, \mu \in \mathbb{R}^d$ *and* $X_1, ..., X_n \overset{i.i.d.}{\sim} P_\theta$ *with* $d > 1$ *and* $n \geq 4d$. *Define* $\cos\beta = |v_1(\sigma^2 I + \mu\mu^T)^T v_1(\widehat{\Sigma}_n)|$. *For any* $0 < \delta < \frac{d-1}{\sqrt{e}}$, *if* $\max\left(\frac{\sigma^2}{\|\mu\|^2}, \frac{\sigma}{\|\mu\|}\right)\sqrt{\frac{\max(d, 8\log\frac{1}{\delta})}{n}} \leq \frac{1}{160}$, *then with probability at least* $1 - 12\delta - 2\exp\left(-\frac{n}{20}\right)$,

$$\sin\beta \leq 14\max\left(\frac{\sigma^2}{\|\mu\|^2}, \frac{\sigma}{\|\mu\|}\right)\sqrt{d}\sqrt{\frac{10}{n}\log\frac{d}{\delta}\max\left(1, \frac{10}{n}\log\frac{d}{\delta}\right)}.$$

**Proposition 7.** *Let* $\theta = (\mu_0 - \mu, \mu_0 + \mu)$, *and for some* $x_0, v \in \mathbb{R}^d$ *with* $\|v\| = 1$, *let* $\widehat{F}(x) = 1$ *if* $x^T v \geq x_0^T v$, *and* $2$ *otherwise. Define* $\cos\beta = |v^T\mu|/\|\mu\|$. *If* $|(x_0 - \mu_0)^T v| \leq \sigma\epsilon_1 + \|\mu\|\epsilon_2$ *for some* $\epsilon_1 \geq 0$ *and* $0 \leq \epsilon_2 \leq \frac{1}{4}$, *and if* $\sin\beta \leq \frac{1}{\sqrt{5}}$, *then*

$$L_\theta(\widehat{F}) \leq \exp\left\{-\frac{1}{2}\max\left(0, \frac{\|\mu\|}{2\sigma} - 2\epsilon_1\right)^2\right\}\left[2\epsilon_1 + \epsilon_2\frac{\|\mu\|}{\sigma} + 2\sin\beta\left(2\sin\beta\frac{\|\mu\|}{\sigma} + 1\right)\right].$$

*Proof.* Let $r = \left|\frac{(x_0 - \mu_0)^T v}{\cos\beta}\right|$. Since the clustering loss is invariant to rotation and translation,

$$L_\theta(\widehat{F}) \leq \frac{1}{2}\int_{-\infty}^{\infty}\frac{1}{\sigma}\phi\left(\frac{x}{\sigma}\right)\left[\Phi\left(\frac{\|\mu\| + |x|\tan\beta + r}{\sigma}\right) - \Phi\left(\frac{\|\mu\| - |x|\tan\beta - r}{\sigma}\right)\right]dx$$

$$\leq \int_{-\infty}^{\infty}\phi(x)\left[\Phi\left(\frac{\|\mu\|}{\sigma}\right) - \Phi\left(\frac{\|\mu\| - r}{\sigma} - |x|\tan\beta\right)\right]dx.$$

Since $\tan\beta \leq \frac{1}{2}$ and $\epsilon_2 \leq \frac{1}{4}$, we have $r \leq 2\sigma\epsilon_1 + 2\|\mu\|\epsilon_2$, and $\Phi\left(\frac{\|\mu\|}{\sigma}\right) - \Phi\left(\frac{\|\mu\|-r}{\sigma}\right) \leq 2\left(\epsilon_1 + \epsilon_2\frac{\|\mu\|}{\sigma}\right)\phi\left(\max\left(0, \frac{\|\mu\|}{2\sigma} - 2\epsilon_1\right)\right)$. Defining $A = \left|\frac{\|\mu\|-r}{\sigma}\right|$,

$$\int_{-\infty}^{\infty}\phi(x)\left[\Phi\left(\frac{\|\mu\| - r}{\sigma}\right) - \Phi\left(\frac{\|\mu\| - r}{\sigma} - |x|\tan\beta\right)\right]dx \leq 2\int_0^{\infty}\int_{A-x\tan\beta}^{A}\phi(x)\phi(y)dydx$$

$$= 2\int_{-A\sin\beta}^{\infty}\int_{A\cos\beta}^{A\cos\beta + (u + A\sin\beta)\tan\beta}\phi(u)\phi(v)dudv \leq 2\phi(A)\tan\beta(A\sin\beta + 1)$$

$$\leq 2\phi\left(\max\left(0, \frac{\|\mu\|}{2\sigma} - 2\epsilon_1\right)\right)\tan\beta\left(\left(2\frac{\|\mu\|}{\sigma} + 2\epsilon_1\right)\sin\beta + 1\right)$$

where we used $u = x\cos\beta - y\sin\beta$ and $v = x\sin\beta + y\cos\beta$ in the second step. The bound now follows easily. $\qquad\square$

*Proof of Theorem 1.* Using Propositions 5 and 6 with $\delta = \frac{1}{\sqrt{n}}$, Proposition 7, and the fact that $(C + x)\exp(-\max(0, x - 4)^2/8) \leq (C + 6)\exp(-\max(0, x - 4)^2/10)$ for all $C, x > 0$,

$$\mathbb{E}_\theta L_\theta(\widehat{F}) \leq 600\max\left(\frac{4\sigma^2}{\lambda^2}, 1\right)\sqrt{\frac{d\log(nd)}{n}}$$

(it is easy to verify that the bounds are decreasing with $\|\mu\|$, so we use $\|\mu\| = \frac{\lambda}{2}$ to bound the supremum). In the $d = 1$ case Proposition 6 need not be applied, since the principal directions agree trivially. The bound for $\frac{\lambda}{\sigma} \geq 2\max(80, 14\sqrt{5d})$ can be shown similarly, using $\delta = \exp\left(-\frac{n}{32}\right)$. $\quad\square$

**Proposition 8.** *Let* $\theta = (\mu_0 - \mu, \mu_0 + \mu)$ *for some* $\mu_0, \mu \in \mathbb{R}^d$ *and* $X_1, ..., X_n \overset{i.i.d.}{\sim} P_\theta$. *For any* $0 < \delta < \frac{1}{\sqrt{e}}$ *such that* $\sqrt{\frac{6\log\frac{1}{\delta}}{n}} \leq \frac{1}{2}$, *with probability at least* $1 - 6d\delta$, *for all* $i \in [d]$,

$$|\widehat{\Sigma}_n(i, i) - (\sigma^2 + \mu(i)^2)| \leq \sigma^2\sqrt{\frac{6\log\frac{1}{\delta}}{n}} + 2\sigma|\mu(i)|\sqrt{\frac{2\log\frac{1}{\delta}}{n}} + (\sigma + |\mu(i)|)^2\frac{2\log\frac{1}{\delta}}{n}.$$

**Proposition 9.** *Let* $\theta = (\mu_0 - \mu, \mu_0 + \mu)$ *for some* $\mu_0, \mu \in \mathbb{R}^d$ *and* $X_1, ..., X_n \overset{i.i.d.}{\sim} P_\theta$. *Define*

$$S(\theta) = \{i \in [d] : \mu(i) \neq 0\} \quad and \quad \widetilde{S}(\theta) = \{i \in [d] : |\mu(i)| \geq 4\sigma\sqrt{\alpha}\}.$$

*Assume that* $n \geq 1$, $d \geq 2$, *and* $\alpha \leq \frac{1}{4}$. *Then* $\widetilde{S}(\theta) \subseteq \widehat{S}_n \subseteq S(\theta)$ *with probability at least* $1 - \frac{6}{n}$.

*Proof.* By Proposition 8, with probability at least $1 - \frac{6}{n}$,

$$|\widehat{\Sigma}_n(i,i) - (\sigma^2 + \mu(i)^2)| \leq \sigma^2 \sqrt{\frac{6\log(nd)}{n}} + 2\sigma|\mu(i)|\sqrt{\frac{2\log(nd)}{n}} + (\sigma + |\mu(i)|)^2 \frac{2\log(nd)}{n}$$

for all $i \in [d]$. Assume the above event holds. If $S(\theta) = [d]$, then of course $\widehat{S}_n \subseteq S(\theta)$. Otherwise, for $i \notin S(\theta)$, we have $(1-\alpha)\sigma^2 \leq \widehat{\Sigma}_n(i,i) \leq (1+\alpha)\sigma^2$, so it is clear that $\widehat{S}_n \subseteq S(\theta)$. The remainder of the proof is trivial if $\widetilde{S}(\theta) = \emptyset$ or $S(\theta) = \emptyset$. Assume otherwise. For any $i \in S(\theta)$,

$$\widehat{\Sigma}_n(i,i) \geq (1-\alpha)\sigma^2 + \left(1 - \frac{2\log(nd)}{n}\right)\mu(i)^2 - 2\alpha\sigma|\mu(i)|.$$

By definition, $|\mu(i)| \geq 4\sigma\sqrt{\alpha}$ for all $i \in \widetilde{S}(\theta)$, so $\frac{(1+\alpha)^2}{1-\alpha}\sigma^2 \leq \widehat{\Sigma}_n(i,i)$ and $i \in \widehat{S}_n$ (we ignore strict equality above as a measure 0 event), i.e. $\widetilde{S}(\theta) \subseteq \widehat{S}_n$, which concludes the proof. $\square$

*Proof of Theorem 3.* Define $S(\theta) = \{i \in [d] : \mu(i) \neq 0\}$ and $\widetilde{S}(\theta) = \{i \in [d] : |\mu(i)| \geq 4\sigma\sqrt{\alpha}\}$. Assume $\widetilde{S}(\theta) \subseteq \widehat{S}_n \subseteq S(\theta)$ (by Proposition 9, this holds with probability at least $1 - \frac{6}{n}$). If $\widetilde{S}(\theta) = \emptyset$, then we simply have $\mathbb{E}_\theta L_\theta(\widehat{F}_n) \leq \frac{1}{2}$.

Assume $\widetilde{S}(\theta) \neq \emptyset$. Let $\cos\widehat{\beta} = |v_1(\widehat{\Sigma}_{\widehat{S}_n})^T v_1(\Sigma)|$, $\cos\widetilde{\beta} = |v_1(\Sigma_{\widehat{S}_n})^T v_1(\Sigma)|$, and $\cos\beta = |v_1(\widehat{\Sigma}_{\widehat{S}_n})^T v_1(\Sigma_{\widehat{S}_n})|$ where $\Sigma = \sigma^2 I + \mu\mu^T$, and for simplicity we define $\widehat{\Sigma}_{\widehat{S}_n}$ and $\Sigma_{\widehat{S}_n}$ to be the same as $\widehat{\Sigma}_n$ and $\Sigma$ in $\widehat{S}_n$, respectively, and 0 elsewhere. Then $\sin\widehat{\beta} \leq \sin\widetilde{\beta} + \sin\beta$, and

$$\sin\widetilde{\beta} = \frac{\|\mu - \mu_{\widehat{S}(\theta)}\|}{\|\mu\|} \leq \frac{\|\mu - \mu_{\widetilde{S}(\theta)}\|}{\|\mu\|} \leq \frac{4\sigma\sqrt{\alpha}\sqrt{|S(\theta)| - |\widetilde{S}(\theta)|}}{\|\mu\|} \leq 8\frac{\sigma\sqrt{s\alpha}}{\lambda}.$$

Using the same argument as the proof of Theorem 1, as long as the above bound is smaller than $\frac{1}{2\sqrt{5}}$,

$$\mathbb{E}_\theta L_\theta(\widehat{F}) \leq 600 \max\left(\frac{\sigma^2}{\left(\frac{\lambda}{2} - 4\sigma\sqrt{s\alpha}\right)^2}, 1\right)\sqrt{\frac{s\log(ns)}{n}} + 104\frac{\sigma\sqrt{s\alpha}}{\lambda} + \frac{3}{n}.$$

Using the fact $L_\theta(\widehat{F}) \leq \frac{1}{2}$ always, and that $\alpha \leq \frac{1}{4}$ implies $\frac{\log(nd)}{n} \leq 1$, the bound follows. $\square$

# 6 Conclusion

We have provided minimax lower and upper bounds for estimating high dimensional mixtures. The bounds show explicitly how the statistical difficulty of the problem depends on dimension $d$, sample size $n$, separation $\lambda$ and sparsity level $s$.

For clarity, we focused on the special case where there are two spherical components with equal mixture weights. In future work, we plan to extend the results to general mixtures of $k$ Gaussians.

One of our motivations for this work is the recent interest in variable selection methods to facilitate clustering in high dimensional problems. Existing methods such as Pan and Shen (2007); Witten and Tibshirani (2010); Raftery and Dean (2006); Sun et al. (2012) and Guo et al. (2010) provide promising numerical evidence that variable selection does improve high dimensional clustering. Our results provide some theoretical basis for this idea.

However, there is a gap between the results in this paper and the above methodology papers. Indeed, as of now, there is no rigorous proof that the methods in those papers outperform a two stage approach where the first stage screens for relevant features and the second stage applies standard clustering methods on the features found in the first stage. We conjecture that there are conditions under which simultaneous feature selection and clustering outperforms a two stage method. Settling this question will require the aforementioned extension of our results to the general mixture case.

## Acknowledgements

This research is supported in part by NSF grants IIS-1116458 and CAREER award IIS-1252412, as well as NSF Grant DMS-0806009 and Air Force Grant FA95500910373.

## Footnotes

[1]We believe our results should also hold in the unequal mixture weight setting without major modifications.

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
