[Supplementary Material]

# Supplementary file for NIPS submission: Minimax Theory for High-dimensional Gaussian Mixtures with Sparse Mean Separation

**Martin Azizyan**
Machine Learning Department
Carnegie Mellon University
Pittsburgh, PA 15213
mazizyan@cs.cmu.edu

**Aarti Singh**
Machine Learning Department
Carnegie Mellon University
Pittsburgh, PA 15213
aarti@cs.cmu.edu

**Larry Wasserman**
Department of Statistics
Carnegie Mellon University
Pittsburgh, PA 15213
larry@stat.cmu.edu

## 1 Notation

For $\theta = (\mu_1, \mu_2) \in \mathbb{R}^{2 \times d}$, define

$$p_\theta(x) = \frac{1}{2} f(x; \mu_1, \sigma^2 I) + \frac{1}{2} f(x; \mu_2, \sigma^2 I),$$

where $f(\cdot; \mu, \Sigma)$ is the density of $\mathcal{N}(\mu, \Sigma)$, $\sigma > 0$ is a fixed constant. Let $P_\theta$ denote the probability measure corresponding to $p_\theta$. We consider two classes $\Theta$ of parameters:

$$\Theta_\lambda = \{(\mu_1, \mu_2) : \|\mu_1 - \mu_2\| \geq \lambda\}$$

$$\Theta_{\lambda,s} = \{(\mu_1, \mu_2) : \|\mu_1 - \mu_2\| \geq \lambda, \ \|\mu_1 - \mu_2\|_0 \leq s\} \subseteq \Theta_\lambda.$$

Throughout this document, $\phi$ and $\Phi$ denote the standard normal density and distribution functions.

For a mixture with parameter $\theta$, the Bayes optimal classification, that is, assignment of a point $x \in \mathbb{R}^d$ to the correct mixture component, is given by the function

$$F_\theta(x) = \underset{i \in \{1,2\}}{\operatorname{argmax}} \ f(x; \mu_i, \sigma^2 I).$$

Given any other candidate assignment function $F : \mathbb{R}^d \to \{1, 2\}$, we define the loss incurred by $F$ as

$$L_\theta(F) = \min_\pi P_\theta(\{x : F_\theta(x) \neq \pi(F(x))\})$$

where the minimum is over all permutations $\pi : \{1, 2\} \to \{1, 2\}$.

For $X_1, \ldots, X_n \overset{\text{i.i.d.}}{\sim} P_\theta$, let $\widehat{\mu}_n$ and $\widehat{\Sigma}_n$ be the mean and covariance of the corresponding empirical distribution.

Also, for a matrix $B$, $v_i(B)$ and $\lambda_i(B)$ are the $i$'th eigenvector and eigenvalue of $B$ (assuming $B$ is symmetric), arranged so that $\lambda_i(B) \geq \lambda_{i+1}(B)$, and $\|B\|_2$ is the spectral norm.

## 2 Upper bounds

### 2.1 Standard concentration bounds

#### 2.1.1 Concentration bounds for estimating the mean

**Proposition 1.** *Let $X \sim \chi_d^2$. Then for any $\epsilon > 0$,*

$$\mathbb{P}(X > (1 + \epsilon)d) \leq \exp\left\{-\frac{d}{2}(\epsilon - \log(1 + \epsilon))\right\}.$$

*If $\epsilon < 1$, then*

$$\mathbb{P}(X < (1-\epsilon)d) \leq \exp\left\{\frac{d}{2}(\epsilon + \log(1-\epsilon))\right\}.$$

*Proof.* Since $\mathbb{E}e^{tX} = (1-2t)^{-\frac{d}{2}}$ for $0 < t < \frac{1}{2}$,

$$\begin{aligned}
\mathbb{P}(X > (1+\epsilon)d) &= \mathbb{P}(e^{tX} > e^{t(1+\epsilon)d}) \\
&\leq e^{-t(1+\epsilon)d}(1-2t)^{-\frac{d}{2}} \\
&= \exp\left[-t(1+\epsilon)d + \frac{d}{2}\log\frac{1}{1-2t}\right].
\end{aligned}$$

To minimize the right hand side, we differentiate the exponent with respect to $t$ to obtain the equation

$$-(1+\epsilon)d + \frac{d}{1-2t} = 0$$

which can be satisfied by setting $t = \frac{1}{2}\left(1 - \frac{1}{1+\epsilon}\right) < \frac{1}{2}$ (it is easy to verify that this is a global minimum). Using this value for $t$, the first bound follows.

Also, for $t > 0$ and $\epsilon < 1$,

$$\begin{aligned}
\mathbb{P}(X < (1-\epsilon)d) &= \mathbb{P}(e^{-tX} > e^{-t(1-\epsilon)d}) \\
&\leq e^{t(1-\epsilon)d}(1+2t)^{-\frac{d}{2}} \\
&= \exp\left[t(1-\epsilon)d - \frac{d}{2}\log(1+2t)\right]
\end{aligned}$$

and setting $t = \frac{1}{2}\left(\frac{1}{1-\epsilon} - 1\right)$,

$$\mathbb{P}(X < (1-\epsilon)d) \leq \exp\left[\frac{d}{2}(\epsilon + \log(1-\epsilon))\right].$$

$\square$

**Proposition 2.** *Let $Z_1, ..., Z_n \overset{i.i.d.}{\sim} \mathcal{N}(0, I_d)$. Then for any $\epsilon > 0$,*

$$\mathbb{P}\left(\left\|\frac{1}{n}\sum_{i=1}^{n}Z_i\right\| \geq \sqrt{\frac{(1+\epsilon)d}{n}}\right) \leq \exp\left\{-\frac{d}{2}(\epsilon - \log(1+\epsilon))\right\}.$$

*Proof.* Using Proposition 1,

$$\begin{aligned}
\mathbb{P}\left(\left\|\frac{1}{n}\sum_{i=1}^{n}Z_i\right\| \geq \sqrt{\frac{(1+\epsilon)d}{n}}\right) &= \mathbb{P}\left(\left\|\frac{1}{\sqrt{n}}\sum_{i=1}^{n}Z_i\right\|^2 \geq (1+\epsilon)d\right) \\
&= \mathbb{P}(X \geq (1+\epsilon)d) \\
&\leq \exp\left\{-\frac{d}{2}(\epsilon - \log(1+\epsilon))\right\}
\end{aligned}$$

where $X \sim \chi_d^2$. $\square$

### 2.1.2 Concentration bounds for estimating principal direction

**Proposition 3.** *Let $Z_1, ..., Z_n \overset{i.i.d.}{\sim} \mathcal{N}(0, I_d)$ and $\delta > 0$. If $n \geq d$ then with probability at least $1 - 3\delta$,*

$$\begin{aligned}
\|\widehat{\Sigma}_n - I_d\|_2 \leq &3\left(1 + \sqrt{\frac{2\log\frac{1}{\delta}}{d}}\right)\sqrt{\frac{d}{n}}\max\left(1, \left(1 + \sqrt{\frac{2\log\frac{1}{\delta}}{d}}\right)\sqrt{\frac{d}{n}}\right) \\
&+ \left(1 + \sqrt{\frac{8\log\frac{1}{\delta}}{d}}\max\left(1, \frac{8\log\frac{1}{\delta}}{d}\right)\right)\frac{d}{n}
\end{aligned}$$

*where $\widehat{\Sigma}_n$ is the empirical covariance of $Z_i$.*

*Proof.* Let $\overline{Z}_n = \frac{1}{n} \sum_{i=1}^n Z_i$. Then

$$\|\widehat{\Sigma}_n - I_d\|_2 = \left\| \frac{1}{n} \sum_{i=1}^n Z_i Z_i^T - I_d - \overline{Z}_n \overline{Z}_n^T \right\|_2$$

$$\leq \left\| \frac{1}{n} \sum_{i=1}^n Z_i Z_i^T - I_d \right\|_2 + \|\overline{Z}_n\|^2.$$

It is well known that for any $\epsilon_1 > 0$,

$$\mathbb{P}\left( \left\| \frac{1}{n} \sum_{i=1}^n Z_i Z_i^T - I_d \right\|_2 \geq 3(1 + \epsilon_1) \sqrt{\frac{d}{n}} \max\left( 1, (1 + \epsilon_1)\sqrt{\frac{d}{n}} \right) \right) \leq 2 \exp\left\{ -\frac{d\epsilon_1^2}{2} \right\}.$$

Using this along with Proposition 2, we have for any $\epsilon_2 > 0$,

$$\mathbb{P}\left( \|\widehat{\Sigma}_n - I_d\|_2 \geq 3(1 + \epsilon_1) \sqrt{\frac{d}{n}} \max\left( 1, (1 + \epsilon_1)\sqrt{\frac{d}{n}} \right) + \frac{(1 + \epsilon_2)d}{n} \right)$$

$$\leq 2 \exp\left\{ -\frac{d\epsilon_1^2}{2} \right\} + \exp\left\{ -\frac{d}{2}(\epsilon_2 - \log(1 + \epsilon_2)) \right\}.$$

Setting $\epsilon_1 = \sqrt{\frac{2 \log \frac{1}{\delta}}{d}}$,

$$\mathbb{P}\left( \|\widehat{\Sigma}_n - I_d\|_2 \geq 3\left( 1 + \sqrt{\frac{2 \log \frac{1}{\delta}}{d}} \right) \sqrt{\frac{d}{n}} \max\left( 1, \left( 1 + \sqrt{\frac{2 \log \frac{1}{\delta}}{d}} \right) \sqrt{\frac{d}{n}} \right) + \frac{(1 + \epsilon_2)d}{n} \right)$$

$$\leq 2\delta + \exp\left\{ -\frac{d}{2}(\epsilon_2 - \log(1 + \epsilon_2)) \right\}$$

$$\leq 2\delta + \exp\left\{ -\frac{d}{8}\epsilon_2 \min(1, \epsilon_2) \right\}$$

and, setting $\epsilon_2 = \sqrt{\frac{8 \log \frac{1}{\delta}}{d} \max\left( 1, \frac{8 \log \frac{1}{\delta}}{d} \right)}$, with probability at least $1 - 3\delta$,

$$\|\widehat{\Sigma}_n - I_d\|_2 \leq 3\left( 1 + \sqrt{\frac{2 \log \frac{1}{\delta}}{d}} \right) \sqrt{\frac{d}{n}} \max\left( 1, \left( 1 + \sqrt{\frac{2 \log \frac{1}{\delta}}{d}} \right) \sqrt{\frac{d}{n}} \right)$$

$$+ \left( 1 + \sqrt{\frac{8 \log \frac{1}{\delta}}{d} \max\left( 1, \frac{8 \log \frac{1}{\delta}}{d} \right)} \right) \frac{d}{n}.$$

$\square$

**Proposition 4.** *Let $X_1, Y_1, ..., X_n, Y_n \overset{i.i.d.}{\sim} \mathcal{N}(0, 1)$. Then for any $\epsilon > 0$,*

$$\mathbb{P}\left( \left| \frac{1}{n} \sum_{i=1}^n X_i Y_i \right| > \frac{\epsilon}{2} \right) \leq 2 \exp\left\{ -\frac{n\epsilon \min(1, \epsilon)}{10} \right\}.$$

*Proof.* Let $Z = XY$ where $X, Y \overset{i.i.d.}{\sim} \mathcal{N}(0, 1)$. Then for any $t$ such that $|t| < 1$,

$$\mathbb{E}e^{tZ} = \frac{1}{\sqrt{1 - t^2}}.$$

So for $0 < t < 1$,

$$
\begin{aligned}
\mathbb{P}\left(\frac{1}{n}\sum_{i=1}^{n} X_i Y_i > \epsilon\right) &= \mathbb{P}\left(\exp\left\{\sum_{i=1}^{n} t X_i Y_i\right\} > \exp(n\epsilon t)\right) \\
&\leq \mathbb{E}\left(\exp\left\{\sum_{i=1}^{n} t X_i Y_i\right\}\right)\exp(-n\epsilon t) \\
&= (\mathbb{E}\exp(t X_i Y_i))^n \exp(-n\epsilon t) \\
&= (1 - t^2)^{-\frac{n}{2}}\exp(-n\epsilon t) \\
&= \exp\left\{-\frac{n}{2}\left(2\epsilon t + \log(1 - t^2)\right)\right\}.
\end{aligned}
$$

The bound is minimized by $t = \frac{1}{2\epsilon}\left(\sqrt{1 + 4\epsilon^2} - 1\right) < 1$, so

$$
\mathbb{P}\left(\frac{1}{n}\sum_{i=1}^{n} X_i Y_i > \epsilon\right) \leq \exp\left\{-\frac{n}{2} h(2\epsilon)\right\}
$$

where

$$
h(u) = \left(\sqrt{1 + u^2} - 1\right) + \log\left(1 - \frac{1}{u^2}\left(\sqrt{1 + u^2} - 1\right)^2\right).
$$

Since $h(u) \geq \frac{u}{5}\min(1, u)$,

$$
\mathbb{P}\left(\frac{1}{n}\sum_{i=1}^{n} X_i Y_i > \epsilon\right) \leq \exp\left\{-\frac{n}{2}\frac{2\epsilon}{5}\min(1, 2\epsilon)\right\}
$$

and the proof is complete by noting that the distribution of $X_i Y_i$ is symmetric. $\square$

## 2.2 Davis–Kahan

**Lemma 1.** *Let $A, E \in \mathbb{R}^{d\times d}$ be symmetric matrices, and $u \in \mathbb{R}^{d-1}$ such that*

$$
u_i = v_{i+1}(A)^T E v_1(A).
$$

*If $\lambda_1(A) - \lambda_2(A) > 0$ and*

$$
\|E\|_2 \leq \frac{\lambda_1(A) - \lambda_2(A)}{5}
$$

*then*

$$
\sqrt{1 - (v_1(A)^T v_1(A + E))^2} \leq \frac{4\|u\|}{\lambda_1(A) - \lambda_2(A)}
$$

*(Corollary 8.1.11 of Golub and Van Loan (1996)).*

## 2.3 Bounding error in estimating the mean

**Proposition 5.** *Let $\theta = (\mu_0 - \mu, \mu_0 + \mu)$ for some $\mu_0, \mu \in \mathbb{R}^d$ and $X_1, ..., X_n \overset{i.i.d.}{\sim} P_\theta$. For any $\delta > 0$,*

$$
\mathbb{P}\left(\|\mu_0 - \widehat{\mu}_n\| \geq \sigma\sqrt{\frac{2\max(d, 8\log\frac{1}{\delta})}{n}} + \|\mu\|\sqrt{\frac{2\log\frac{1}{\delta}}{n}}\right) \leq 3\delta.
$$

*Proof.* Let $Z_1, ..., Z_n \overset{\text{i.i.d.}}{\sim} \mathcal{N}(0, I)$ and $Y_1, ..., Y_n$ i.i.d. such that $\mathbb{P}(Y_i = -1) = \mathbb{P}(Y_i = 1) = \frac{1}{2}$. Then for any $\epsilon_1, \epsilon_2 > 0$,

$$\mathbb{P}\left(\|\mu_0 - \widehat{\mu}_n\| \geq \sigma\sqrt{\frac{(1+\epsilon_1)d}{n}} + \|\mu\|\epsilon_2\right)$$

$$= \mathbb{P}\left(\left\|\mu_0 - \frac{1}{n}\sum_{i=1}^{d}(\sigma Z_i + \mu_0 + \mu Y_i)\right\| \geq \sigma\sqrt{\frac{(1+\epsilon_1)d}{n}} + \|\mu\|\epsilon_2\right)$$

$$= \mathbb{P}\left(\left\|\sigma\frac{1}{n}\sum_{i=1}^{d}Z_i + \mu\frac{1}{n}\sum_{i=1}^{d}Y_i\right\| \geq \sigma\sqrt{\frac{(1+\epsilon_1)d}{n}} + \|\mu\|\epsilon_2\right)$$

$$\leq \mathbb{P}\left(\sigma\left\|\frac{1}{n}\sum_{i=1}^{d}Z_i\right\| + \|\mu\|\left|\frac{1}{n}\sum_{i=1}^{d}Y_i\right| \geq \sigma\sqrt{\frac{(1+\epsilon_1)d}{n}} + \|\mu\|\epsilon_2\right)$$

$$\leq \mathbb{P}\left(\left\|\frac{1}{n}\sum_{i=1}^{d}Z_i\right\| \geq \sqrt{\frac{(1+\epsilon_1)d}{n}}\right) + \mathbb{P}\left(\left|\frac{1}{n}\sum_{i=1}^{d}Y_i\right| \geq \epsilon_2\right)$$

$$\leq \exp\left\{-\frac{d}{2}(\epsilon_1 - \log(1+\epsilon_1))\right\} + 2\exp\left\{-\frac{n\epsilon_2^2}{2}\right\}$$

where the last step is using Hoeffding's inequality and Proposition 2. Setting $\epsilon_2 = \sqrt{\frac{2\log\frac{1}{\delta}}{n}}$,

$$\mathbb{P}\left(\|\mu_0 - \widehat{\mu}_n\| \geq \sigma\sqrt{\frac{(1+\epsilon_1)d}{n}} + \|\mu\|\sqrt{\frac{2\log\frac{1}{\delta}}{n}}\right)$$

$$\leq \exp\left\{-\frac{d}{2}(\epsilon_1 - \log(1+\epsilon_1))\right\} + 2\delta.$$

Since $\epsilon_1 - \log(1+\epsilon_1) \geq \frac{\epsilon_1}{4}\min(1, \epsilon_1)$,

$$\exp\left\{-\frac{d}{2}(\epsilon_1 - \log(1+\epsilon_1))\right\} \leq \exp\left\{-\frac{d}{8}\epsilon_1\min(1, \epsilon_1)\right\}.$$

Setting

$$\epsilon_1 = \sqrt{\frac{8\log\frac{1}{\delta}}{d}\max\left(1, \frac{8\log\frac{1}{\delta}}{d}\right)},$$

we have

$$\mathbb{P}\left(\|\mu_0 - \widehat{\mu}_n\| \geq \sigma\sqrt{\frac{d}{n}\left(1 + \sqrt{\frac{8\log\frac{1}{\delta}}{d}\max\left(1, \frac{8\log\frac{1}{\delta}}{d}\right)}\right)} + \|\mu\|\sqrt{\frac{2\log\frac{1}{\delta}}{n}}\right) \leq 3\delta$$

and the bound follows. $\qquad\square$

## 2.4 Bounding error in estimating principal direction

**Proposition 6.** *Let $\theta = (\mu_0 - \mu, \mu_0 + \mu)$ for some $\mu_0, \mu \in \mathbb{R}^d$ and $X_1, ..., X_n \overset{i.i.d.}{\sim} P_\theta$. If $n \geq d$ then for any $\delta, \delta_1 > 0$, with probability at least $1 - 5\delta - 2\delta_1$,*

$$\|\widehat{\Sigma}_n - (\sigma^2 I_d + \mu\mu^T)\|_2$$

$$\leq 3\sigma^2 \left(1 + \sqrt{\frac{2\log\frac{1}{\delta}}{d}}\right) \sqrt{\frac{d}{n}} \max\left(1, \left(1 + \sqrt{\frac{2\log\frac{1}{\delta}}{d}}\right)\sqrt{\frac{d}{n}}\right)$$

$$+ \sigma^2 \left(1 + \sqrt{\frac{8\log\frac{1}{\delta}}{d} \max\left(1, \frac{8\log\frac{1}{\delta}}{d}\right)}\right) \frac{d}{n}$$

$$+ 4\sigma\|\mu\| \sqrt{\left(1 + \sqrt{\frac{8\log\frac{1}{\delta}}{d} \max\left(1, \frac{8\log\frac{1}{\delta}}{d}\right)}\right) \frac{d}{n}} + \frac{2\|\mu\|^2 \log\frac{1}{\delta_1}}{n}$$

*where $\widehat{\Sigma}_n$ is the empirical covariance of $X_i$.*

*Proof.* We can express $X_i$ as $X_i = \sigma Z_i + \mu Y_i + \mu_0$ where $Z_1, ..., Z_n \overset{i.i.d.}{\sim} \mathcal{N}(0, I_d)$ and $Y_1, ..., Y_n$ i.i.d. such that $\mathbb{P}(Y_i = -1) = \mathbb{P}(Y_i = 1) = \frac{1}{2}$. Then

$$\widehat{\Sigma}_n - (\sigma^2 I_d + \mu\mu^T) = \sigma^2(\widehat{\Sigma}_n^Z - I_d) - \mu\mu^T \overline{Y}^2$$

$$+ \sigma\left(\frac{1}{n}\sum_{i=1}^n Y_i Z_i - \overline{Y}\,\overline{Z}\right)\mu^T$$

$$+ \sigma\mu\left(\frac{1}{n}\sum_{i=1}^n Y_i Z_i - \overline{Y}\,\overline{Z}\right)^T$$

where $\widehat{\Sigma}_n^Z$ is the empirical covariance of $Z_i$ and $\overline{Y}$ and $\overline{Z}$ are the empirical means of $Y_i$ and $Z_i$. So

$$\|\widehat{\Sigma}_n - (\sigma^2 I_d + \mu\mu^T)\|_2 \leq \sigma^2\|\widehat{\Sigma}_n^Z - I_d\|_2 + \|\mu\|^2 \overline{Y}^2$$

$$+ 2\sigma\|\mu\| \left(\left\|\frac{1}{n}\sum_{i=1}^n Y_i Z_i\right\| + |\overline{Y}|\|\overline{Z}\|\right).$$

By Hoeffding's inequality,

$$\mathbb{P}\left(\|\mu\|^2 \overline{Y}^2 \geq \frac{2\|\mu\|^2 \log\frac{1}{\delta_1}}{n}\right) \leq 2\delta_1.$$

Since $|\overline{Y}| \leq 1$ and since $Y_i Z_i$ has the same distribution as $Z_i$, by Proposition 2, for any $\epsilon > 0$,

$$\mathbb{P}\left(2\sigma\|\mu\|\left(\left\|\frac{1}{n}\sum_{i=1}^n Y_i Z_i\right\| + |\overline{Y}|\|\overline{Z}\|\right) \geq 4\sigma\|\mu\|\sqrt{\frac{(1+\epsilon)d}{n}}\right)$$

$$\leq 2\exp\left\{-\frac{d}{2}(\epsilon - \log(1+\epsilon))\right\} \leq 2\exp\left\{-\frac{d}{8}\epsilon\min(1,\epsilon)\right\}.$$

Setting

$$\epsilon = \sqrt{\frac{8\log\frac{1}{\delta}}{d} \max\left(1, \frac{8\log\frac{1}{\delta}}{d}\right)}$$

we have

$$\mathbb{P}\left(2\sigma\|\mu\|\left(\left\|\frac{1}{n}\sum_{i=1}^n Y_i Z_i\right\| + |\overline{Y}|\|\overline{Z}\|\right) \geq 4\sigma\|\mu\|\sqrt{\left(1 + \sqrt{\frac{8\log\frac{1}{\delta}}{d}\max\left(1, \frac{8\log\frac{1}{\delta}}{d}\right)}\right)\frac{d}{n}}\right)$$

$$\leq 2\delta.$$

Finally, by Proposition 3, with probability at least $1 - 3\delta$,

$$
\sigma^2 \|\widehat{\Sigma}_n^Z - I_d\|_2 \leq 3\sigma^2 \left(1 + \sqrt{\frac{2\log\frac{1}{\delta}}{d}}\right) \sqrt{\frac{d}{n}} \max\left(1, \left(1 + \sqrt{\frac{2\log\frac{1}{\delta}}{d}}\right) \sqrt{\frac{d}{n}}\right)
$$

$$
+ \sigma^2 \left(1 + \sqrt{\frac{8\log\frac{1}{\delta}}{d} \max\left(1, \frac{8\log\frac{1}{\delta}}{d}\right)}\right) \frac{d}{n}
$$

and we complete the proof by combining the three bounds. $\qquad \square$

**Proposition 7.** *Let $\theta = (\mu_0 - \mu, \mu_0 + \mu)$ for some $\mu_0, \mu \in \mathbb{R}^d$ and $X_1, ..., X_n \overset{i.i.d.}{\sim} P_\theta$. If $n \geq d > 1$ then for any $0 < \delta \leq \frac{1}{\sqrt{e}}$ and $i \in [2..d]$, with probability at least $1 - 7\delta$,*

$$
\left| v_i(\sigma^2 I + \mu\mu^T)^T (\widehat{\Sigma}_n - (\sigma^2 I + \mu\mu^T)) v_1(\sigma^2 I + \mu\mu^T) \right|
$$

$$
\leq \sigma^2 \frac{1}{2} \sqrt{\frac{10\log\frac{1}{\delta}}{n} \max\left(1, \frac{10\log\frac{1}{\delta}}{n}\right)} + \sigma\|\mu\| \sqrt{\frac{2\log\frac{1}{\delta}}{n}} + (\sigma^2 + \sigma\|\mu\|) \frac{2\log\frac{1}{\delta}}{n}.
$$

*Proof.* Let $Z_1, W_1, ..., Z_n, W_n \overset{i.i.d.}{\sim} \mathcal{N}(0,1)$ and $Y_1, ..., Y_n$ i.i.d. such that $\mathbb{P}(Y_i = -1) = \mathbb{P}(Y_i = 1) = \frac{1}{2}$. It is easy to see that the quantity of interest is equal in distribution to

$$
\left| \frac{1}{n} \sum_{j=1}^n (\sigma Z_i - \sigma\overline{Z})(\sigma W_i - \sigma\overline{W} + \|\mu\|Y_i - \|\mu\|\overline{Y}) \right|
$$

where $\overline{Z}, \overline{W}, \overline{Y}$ are the respective empirical means. Moreover,

$$
\left| \frac{1}{n} \sum_{j=1}^n (\sigma Z_i - \sigma\overline{Z})(\sigma W_i - \sigma\overline{W} + \|\mu\|Y_i - \|\mu\|\overline{Y}) \right|
$$

$$
\leq \sigma^2 \left| \frac{1}{n} \sum_{j=1}^n Z_i W_i \right| + \sigma^2 |\overline{Z}| |\overline{W}| + \sigma\|\mu\| \left| \frac{1}{n} \sum_{j=1}^n Z_i Y_i \right| + \sigma\|\mu\| |\overline{Z}| |\overline{Y}|.
$$

From Proposition 4, we have

$$
\mathbb{P}\left( \left| \frac{1}{n} \sum_{i=1}^n Z_i W_i \right| > \frac{1}{2} \sqrt{\frac{10\log\frac{1}{\delta}}{n} \max\left(1, \frac{10\log\frac{1}{\delta}}{n}\right)} \right) \leq 2\delta;
$$

using Hoeffding's inequality,

$$
\mathbb{P}\left( |\overline{Y}| \geq \sqrt{\frac{2\log\frac{1}{\delta}}{n}} \right) \leq 2\delta;
$$

and using the Gaussian tail bound, for $\delta \leq \frac{1}{\sqrt{e}}$,

$$
\mathbb{P}\left( |\overline{Z}| \geq \sqrt{\frac{2\log\frac{1}{\delta}}{n}} \right) \leq \delta
$$

and the final result follows easily. $\qquad \square$

**Proposition 8.** *Let $\theta = (\mu_0 - \mu, \mu_0 + \mu)$ for some $\mu_0, \mu \in \mathbb{R}^d$ and $X_1, ..., X_n \overset{i.i.d.}{\sim} P_\theta$ with $d > 1$ and $n \geq 4d$. For any $0 < \delta < \frac{d-1}{\sqrt{e}}$, if*

$$
\max\left(\frac{\sigma^2}{\|\mu\|^2}, \frac{\sigma}{\|\mu\|}\right) \sqrt{\frac{\max(d, 8\log\frac{1}{\delta})}{n}} \leq \frac{1}{160}
$$

*then with probability at least* $1 - 12\delta - 2\exp\left(-\frac{n}{20}\right)$,

$$\sqrt{1 - (v_1(\sigma^2 I + \mu\mu^T)^T v_1(\widehat{\Sigma}_n))^2} \leq 14 \max\left(\frac{\sigma^2}{\|\mu\|^2}, \frac{\sigma}{\|\mu\|}\right) \sqrt{d} \sqrt{\frac{10\log\frac{d}{\delta}}{n} \max\left(1, \frac{10\log\frac{d}{\delta}}{n}\right)}.$$

*Proof.* By Proposition 6 (with $\delta_1 = \exp\left(-\frac{n}{20}\right)$), Proposition 7 (with $\delta_2 = \frac{\delta}{d-1}$), and Lemma 1, with probability at least $1 - 12\delta - 2\exp\left(-\frac{n}{20}\right)$,

$$\sqrt{1 - (v_1(\sigma^2 I + \mu\mu^T)^T v_1(\widehat{\Sigma}_n))^2}$$

$$\leq \frac{4\sqrt{d-1}}{\|\mu\|^2}\left[\sigma^2 \frac{1}{2}\sqrt{\frac{10\log\frac{d-1}{\delta}}{n}\max\left(1, \frac{10\log\frac{d-1}{\delta}}{n}\right)} + \sigma\|\mu\|\sqrt{\frac{2\log\frac{d-1}{\delta}}{n}} + (\sigma^2 + \sigma\|\mu\|)\frac{2\log\frac{d-1}{\delta}}{n}\right]$$

and the result follows after some simplifications.

$\square$

## 2.5 General result relating error in estimating mean and principal direction to clustering loss

**Proposition 9.** *Let* $\theta = (\mu_0 - \mu, \mu_0 + \mu)$ *and let*

$$\widehat{F}(x) = \begin{cases} 1 & \text{if } x^T v \geq x_0^T v \\ 2 & \text{otherwise} \end{cases}$$

*for some* $x_0, v \in \mathbb{R}^d$, *with* $\|v\| = 1$. *Define* $\cos\beta = |v^T\mu|/\|\mu\|$. *If* $|(x_0 - \mu_0)^T v| \leq \sigma\epsilon_1 + \|\mu\|\epsilon_2$ *for some* $\epsilon_1 \geq 0$ *and* $0 \leq \epsilon_2 \leq \frac{1}{4}$, *and if* $\sin\beta \leq \frac{1}{\sqrt{5}}$, *then*

$$L_\theta(\widehat{F}) \leq \exp\left\{-\frac{1}{2}\max\left(0, \frac{\|\mu\|}{2\sigma} - 2\epsilon_1\right)^2\right\}\left[2\epsilon_1 + \epsilon_2\frac{\|\mu\|}{\sigma} + 2\sin\beta\left(2\sin\beta\frac{\|\mu\|}{\sigma} + 1\right)\right].$$

*Proof.*

$$L_\theta(\widehat{F}) = \min_\pi P_\theta(\{x : F_\theta(x) \neq \pi(\widehat{F}(x))\})$$
$$= \min\left\{P_\theta[\{x : ((x - \mu_0)^T\mu)((x - x_0)^T v) \geq 0\}],\right.$$
$$\left. P_\theta[\{x : ((x - \mu_0)^T\mu)((x - x_0)^T v) \leq 0\}]\right\}.$$

WLOG assume $v^T\mu \geq 0$ (otherwise we can simply replace $v$ with $-v$, which does not affect the bound). Then

$$L_\theta(\widehat{F}) = P_\theta[\{x : ((x - \mu_0)^T\mu)((x - x_0)^T v) \leq 0\}]$$
$$= P_\theta[\{x : ((x - \mu_0)^T\mu)((x - \mu_0)^T v - (x_0 - \mu_0)^T v) \leq 0\}]$$
$$= P_\theta\left[\left\{x : \left((x - \mu_0)^T\frac{\mu}{\|\mu\|}\right)((x - \mu_0)^T v - (x_0 - \mu_0)^T v) \leq 0\right\}\right].$$

Define

$$\breve{\mu} = \frac{\mu}{\|\mu\|},$$
$$\breve{x} = (x - \mu_0)^T\breve{\mu},$$

and

$$\breve{y} = (x - \mu_0)^T\frac{v - \breve{\mu}\breve{\mu}^T v}{\|v - \breve{\mu}\breve{\mu}^T v\|} \equiv (x - \mu_0)^T\frac{v - \breve{\mu}\breve{\mu}^T v}{\sin\beta}$$

so that

$$L_\theta(\widehat{F}) = P_\theta\left[\left\{x : \breve{x}\left(\breve{y}\sin\beta + \breve{x}\cos\beta - (x_0 - \mu_0)^T v\right) \leq 0\right\}\right]$$
$$= P_\theta\left[\{x : \min(0, B(\breve{y})) \leq \breve{x} \leq \max(0, B(\breve{y}))\}\right]$$

where
$$B(\breve{y}) = \frac{(x_0 - \mu_0)^T v}{\cos\beta} - \breve{y}\tan\beta.$$

Since $\breve{x}$ and $\breve{y}$ are projections of $x - \mu_0$ onto orthogonal unit vectors, and since $\breve{x}$ is exactly the component of $x - \mu_0$ that lies in the direction of $\mu$, we can integrate out all other directions and obtain

$$L_\theta(\widehat{F}) = \int_{-\infty}^{\infty} \phi_\sigma(\breve{y}) \int_{\min(0,B(\breve{y}))}^{\max(0,B(\breve{y}))} \left(\frac{1}{2}\phi_\sigma(\breve{x} + \|\mu\|) + \frac{1}{2}\phi_\sigma(\breve{x} - \|\mu\|)\right) d\breve{x}\, d\breve{y}$$

where $\phi_\sigma$ is the density of $\mathcal{N}(0,\sigma^2)$. But,

$$\int_{\min(0,B(\breve{y}))}^{\max(0,B(\breve{y}))} \left(\frac{1}{2}\phi_\sigma(\breve{x} + \|\mu\|) + \frac{1}{2}\phi_\sigma(\breve{x} - \|\mu\|)\right) d\breve{x}$$

$$= \frac{1}{2}\int_{\min(0,B(\breve{y}))}^{\max(0,B(\breve{y}))} \phi_\sigma(\breve{x} + \|\mu\|) d\breve{x} + \frac{1}{2}\int_{\min(0,B(\breve{y}))}^{\max(0,B(\breve{y}))} \phi_\sigma(\breve{x} - \|\mu\|) d\breve{x}$$

$$= \frac{1}{2}\left(\Phi\left(\frac{\max(0,B(\breve{y})) + \|\mu\|}{\sigma}\right) - \Phi\left(\frac{\min(0,B(\breve{y})) + \|\mu\|}{\sigma}\right)\right)$$

$$+ \frac{1}{2}\left(-\Phi\left(\frac{-\max(0,B(\breve{y})) + \|\mu\|}{\sigma}\right) + \Phi\left(\frac{-\min(0,B(\breve{y})) + \|\mu\|}{\sigma}\right)\right)$$

$$= \frac{1}{2}\left(\Phi\left(\frac{\|\mu\| + |B(\breve{y})|}{\sigma}\right) - \Phi\left(\frac{\|\mu\| - |B(\breve{y})|}{\sigma}\right)\right).$$

Since the above quantity is increasing in $|B(\breve{y})|$, and since $|B(\breve{y})| \le |\breve{y}|\tan\beta + r$ where

$$r = \left|\frac{(x_0 - \mu_0)^T v}{\cos\beta}\right|,$$

we have that, replacing $\breve{y}$ by $x$,

$$L_\theta(\widehat{F}) \le \frac{1}{2}\int_{-\infty}^{\infty} \frac{1}{\sigma}\phi\left(\frac{x}{\sigma}\right)\left[\Phi\left(\frac{\|\mu\| + |x|\tan\beta + r}{\sigma}\right) - \Phi\left(\frac{\|\mu\| - |x|\tan\beta - r}{\sigma}\right)\right] dx$$

$$\le \int_{-\infty}^{\infty} \frac{1}{\sigma}\phi\left(\frac{x}{\sigma}\right)\left[\Phi\left(\frac{\|\mu\|}{\sigma}\right) - \Phi\left(\frac{\|\mu\| - |x|\tan\beta - r}{\sigma}\right)\right] dx$$

$$= \int_{-\infty}^{\infty} \phi(x)\left[\Phi\left(\frac{\|\mu\| - r}{\sigma}\right) - \Phi\left(\frac{\|\mu\| - r}{\sigma} - |x|\tan\beta\right)\right] dx$$

$$+ \left[\Phi\left(\frac{\|\mu\|}{\sigma}\right) - \Phi\left(\frac{\|\mu\| - r}{\sigma}\right)\right].$$

Since $\tan\beta \le \frac{1}{2}$, we have that $r \le 2|(x_0 - \mu_0)^T v| \le 2\sigma\epsilon_1 + 2\|\mu\|\epsilon_2$ and

$$\Phi\left(\frac{\|\mu\|}{\sigma}\right) - \Phi\left(\frac{\|\mu\| - r}{\sigma}\right) \le \frac{r}{\sigma}\phi\left(\max\left(0, \frac{\|\mu\| - r}{\sigma}\right)\right)$$

$$\le \left(2\epsilon_1 + 2\epsilon_2\frac{\|\mu\|}{\sigma}\right)\phi\left(\max\left(0, (1 - 2\epsilon_2)\frac{\|\mu\|}{\sigma} - 2\epsilon_1\right)\right),$$

and since $\epsilon_2 \le \frac{1}{4}$,

$$\Phi\left(\frac{\|\mu\|}{\sigma}\right) - \Phi\left(\frac{\|\mu\| - r}{\sigma}\right) \le 2\left(\epsilon_1 + \epsilon_2\frac{\|\mu\|}{\sigma}\right)\phi\left(\max\left(0, \frac{\|\mu\|}{2\sigma} - 2\epsilon_1\right)\right).$$

Defining $A = \left| \frac{\|\mu\| - r}{\sigma} \right|$,

$$\int_{-\infty}^{\infty} \phi(x) \left[ \Phi\left( \frac{\|\mu\| - r}{\sigma} \right) - \Phi\left( \frac{\|\mu\| - r}{\sigma} - |x| \tan \beta \right) \right] dx$$

$$\leq 2 \int_0^{\infty} \int_{A - x \tan \beta}^{A} \phi(x)\phi(y) dy dx = 2 \int_{-A \sin \beta}^{\infty} \int_{A \cos \beta}^{A \cos \beta + (x + A \sin \beta) \tan \beta} \phi(x)\phi(y) dy dx$$

$$\leq 2\phi(A \cos \beta) \tan \beta \int_{-A \sin \beta}^{\infty} (x + A \sin \beta)\phi(x) dx$$

$$= 2\phi(A \cos \beta) \tan \beta \left( A \sin \beta \Phi(A \sin \beta) + \phi(A \sin \beta) \right)$$

$$\leq 2\phi(A) \tan \beta \left( A \sin \beta + 1 \right)$$

$$\leq 2\phi\left( \max\left( 0, \frac{\|\mu\| - r}{\sigma} \right) \right) \tan \beta \left( \left( \frac{\|\mu\| + r}{\sigma} \right) \sin \beta + 1 \right)$$

and

$$\int_{-\infty}^{\infty} \phi(x) \left[ \Phi\left( \frac{\|\mu\| - r}{\sigma} \right) - \Phi\left( \frac{\|\mu\| - r}{\sigma} - |x| \tan \beta \right) \right] dx$$

$$\leq 2\phi\left( \max\left( 0, \frac{\|\mu\|}{2\sigma} - 2\epsilon_1 \right) \right) \tan \beta \left( \left( 2\frac{\|\mu\|}{\sigma} + 2\epsilon_1 \right) \sin \beta + 1 \right).$$

So we have that

$$L_\theta(\widehat{F}) \leq 2 \left( \epsilon_1 + \epsilon_2 \frac{\|\mu\|}{\sigma} \right) \phi\left( \max\left( 0, \frac{\|\mu\|}{2\sigma} - 2\epsilon_1 \right) \right)$$

$$+ 2\phi\left( \max\left( 0, \frac{\|\mu\|}{2\sigma} - 2\epsilon_1 \right) \right) \tan \beta \left( \left( 2\frac{\|\mu\|}{\sigma} + 2\epsilon_1 \right) \sin \beta + 1 \right)$$

$$\leq \phi\left( \max\left( 0, \frac{\|\mu\|}{2\sigma} - 2\epsilon_1 \right) \right) \times$$

$$\times \left[ 2\epsilon_1 + 2\epsilon_2 \frac{\|\mu\|}{\sigma} + 4 \sin \beta \tan \beta \frac{\|\mu\|}{\sigma} + 4\epsilon_1 \sin \beta \tan \beta + 2 \tan \beta \right]$$

$$\leq \exp\left\{ -\frac{1}{2} \max\left( 0, \frac{\|\mu\|}{2\sigma} - 2\epsilon_1 \right)^2 \right\} \left[ 2\epsilon_1 + \epsilon_2 \frac{\|\mu\|}{\sigma} + \tan \beta \left( 2 \sin \beta \frac{\|\mu\|}{\sigma} + 1 \right) \right].$$

$\square$

## 2.6  Non-sparse upper bound

**Theorem 1.** *For any $\theta \in \Theta_\lambda$ and $X_1, ..., X_n \overset{i.i.d.}{\sim} P_\theta$, let*

$$\widehat{F}(x) = \begin{cases} 1 & \text{if } x^T v_1(\widehat{\Sigma}_n) \geq \widehat{\mu}_n^T v_1(\widehat{\Sigma}_n) \\ 2 & \text{otherwise,} \end{cases}$$

*and let $n \geq \max(68, 4d)$, $d \geq 1$.*

*Then*

$$\sup_{\theta \in \Theta_\lambda} \mathbb{E} L_\theta(\widehat{F}) \leq 600 \max\left( \frac{4\sigma^2}{\lambda^2}, 1 \right) \sqrt{\frac{d \log(nd)}{n}}.$$

*Furthermore, if $\frac{\lambda}{\sigma} \geq 2 \max(80, 14\sqrt{5d})$, then*

$$\sup_{\theta \in \Theta_\lambda} \mathbb{E} L_\theta(\widehat{F}) \leq 17 \exp\left( -\frac{n}{32} \right) + 9 \exp\left( -\frac{\lambda^2}{80\sigma^2} \right).$$

*Proof.* Using Propositions 5 and 8 with $\delta = \frac{1}{\sqrt{n}}$, Proposition 9, and the fact that $(C + x)\exp(-\max(0, x-4)^2/8) \leq (C+6)\exp(-\max(0, x-4)^2/10)$ for all $C, x > 0$,

$$\mathbb{E}L_\theta(\widehat{F}) \leq 600 \max\left(\frac{4\sigma^2}{\lambda^2}, 1\right)\sqrt{\frac{d\log(nd)}{n}}$$

(it is easy to verify that the bounds are decreasing with $\|\mu\|$, so we use $\|\mu\| = \frac{\lambda}{2}$ to bound the supremum). Note that the $d = 1$ case must be handled separately, but results in a bound that agrees with the above.

Also, when $\frac{\lambda}{\sigma} \geq 2\max(80, 14\sqrt{5d})$, using $\delta = \exp\left(-\frac{n}{32}\right)$,

$$\mathbb{E}L_\theta(\widehat{F}) \leq 17\exp\left(-\frac{n}{32}\right) + 9\exp\left(-\frac{\lambda^2}{80\sigma^2}\right).$$

$\square$

## 2.7 Estimating the support in the sparse case

**Proposition 10.** *Let* $\theta = (\mu_0 - \mu, \mu_0 + \mu)$ *for some* $\mu_0, \mu \in \mathbb{R}^d$ *and* $X_1, ..., X_n \overset{i.i.d.}{\sim} P_\theta$. *For any* $0 < \delta < \frac{1}{\sqrt{e}}$ *such that* $\sqrt{\frac{6\log\frac{1}{\delta}}{n}} \leq \frac{1}{2}$, *with probability at least* $1 - 6d\delta$,

$$|\widehat{\Sigma}_n(i,i) - (\sigma^2 + \mu(i)^2)| \leq \sigma^2\sqrt{\frac{6\log\frac{1}{\delta}}{n}} + 2\sigma|\mu(i)|\sqrt{\frac{2\log\frac{1}{\delta}}{n}} + (\sigma + |\mu(i)|)^2\frac{2\log\frac{1}{\delta}}{n}$$

*for all* $i \in [d]$.

*Proof.* Consider any $i \in [d]$. Let $Z_1, ..., Z_n \overset{i.i.d.}{\sim} \mathcal{N}(0, 1)$ and $Y_1, ..., Y_n$ i.i.d. such that $\mathbb{P}(Y_j = -1) = \mathbb{P}(Y_j = 1) = \frac{1}{2}$. Then $\widehat{\Sigma}_n(i,i)$ is equal in distribution to

$$\frac{1}{n}\sum_{j=1}^n(\sigma Z_j + \mu(i)Y_j - \sigma\overline{Z} - \mu(i)\overline{Y})^2$$

where $\overline{Z}$ and $\overline{Y}$ are the respective empirical means, and

$$\frac{1}{n}\sum_{j=1}^n(\sigma Z_j + \mu(i)Y_j - \sigma\overline{Z} - \mu(i)\overline{Y})^2 = \frac{1}{n}\sum_{j=1}^n(\sigma Z_j + \mu(i)Y_j)^2 - (\sigma\overline{Z} + \mu(i)\overline{Y})^2$$

$$= \sigma^2\frac{1}{n}\sum_{j=1}^n Z_j^2 + \mu(i)^2 + 2\sigma\mu(i)\frac{1}{n}\sum_{j=1}^n Z_j Y_j$$

$$- \sigma^2\overline{Z}^2 - \mu(i)^2\overline{Y}^2 - 2\sigma\mu(i)\overline{ZY}$$

So, by Hoeffding's inequality, a Gaussian tail bound, and Proposition 1, we have that for any $0 < \delta < \frac{1}{\sqrt{e}}$, with probability at least $1 - 6\delta$,

$$|\widehat{\Sigma}_n(i,i) - (\sigma^2 + \mu(i)^2)| \leq \sigma^2\sqrt{\frac{6\log\frac{1}{\delta}}{n}} + 2\sigma|\mu(i)|\sqrt{\frac{2\log\frac{1}{\delta}}{n}} + (\sigma + |\mu(i)|)^2\frac{2\log\frac{1}{\delta}}{n}$$

where we have used the fact that for $\epsilon \in (0, 0.5]$,

$$\max\{-\epsilon + \log(1+\epsilon), \epsilon + \log(1-\epsilon)\} \leq -\frac{\epsilon^2}{3}$$

and the result follows easily. $\square$

**Proposition 11.** *Let* $\theta = (\mu_0 - \mu, \mu_0 + \mu)$ *for some* $\mu_0, \mu \in \mathbb{R}^d$ *and* $X_1, ..., X_n \overset{i.i.d.}{\sim} P_\theta$. *Define*

$$S(\theta) = \{i \in [d] : \mu(i) \neq 0\},$$

$$\alpha = \sqrt{\frac{6\log(nd)}{n}} + \frac{2\log(nd)}{n},$$

$$\widetilde{S}(\theta) = \{i \in [d] : |\mu(i)| \geq 4\sigma\sqrt{\alpha}\},$$

$$\widehat{\tau}_n = \frac{1+\alpha}{1-\alpha} \min_{i \in [d]} \widehat{\Sigma}_n(i,i),$$

*and*

$$\widehat{S}_n = \{i \in [d] : \widehat{\Sigma}_n(i,i) > \widehat{\tau}_n\}.$$

*Assume that* $n \geq 1$, $d \geq 2$, *and* $\alpha \leq \frac{1}{4}$. *Then* $\widetilde{S}(\theta) \subseteq \widehat{S}_n \subseteq S(\theta)$ *with probability at least* $1 - \frac{6}{n}$.

*Proof.* By Proposition 10, with probability at least $1 - \frac{6}{n}$,

$$|\widehat{\Sigma}_n(i,i) - (\sigma^2 + \mu(i)^2)| \leq \sigma^2\sqrt{\frac{6\log(nd)}{n}} + 2\sigma|\mu(i)|\sqrt{\frac{2\log(nd)}{n}} + (\sigma + |\mu(i)|)^2\frac{2\log(nd)}{n}$$

for all $i \in [d]$. Assume the above event holds. If $S(\theta) = [d]$, then of course $\widehat{S}_n \subseteq S(\theta)$. Otherwise, for $i \notin S(\theta)$,

$$(1-\alpha)\sigma^2 \leq \widehat{\Sigma}_n(i,i) \leq (1+\alpha)\sigma^2$$

so it is clear that $\widehat{S}_n \subseteq S(\theta)$.

The remainder of the proof is trivial if $\widetilde{S}(\theta) = \emptyset$ or $S(\theta) = \emptyset$. Assume otherwise. For any $i \in S(\theta)$,

$$\widehat{\Sigma}_n(i,i) \geq (1-\alpha)\sigma^2 + \mu(i)^2 - 2\sigma|\mu(i)|\sqrt{\frac{2\log(nd)}{n}} - 2\sigma|\mu(i)|\frac{2\log(nd)}{n} - \mu(i)^2\frac{2\log(nd)}{n}$$

$$\geq (1-\alpha)\sigma^2 + \left(1 - \frac{2\log(nd)}{n}\right)\mu(i)^2 - 2\alpha\sigma|\mu(i)|.$$

By definition, $|\mu(i)| \geq 4\sigma\sqrt{\alpha}$ for all $i \in \widetilde{S}(\theta)$, so

$$\frac{(1+\alpha)^2}{1-\alpha}\sigma^2 \leq (1-\alpha)\sigma^2 + \left(1 - \frac{2\log(nd)}{n}\right)\mu(i)^2 - 2\alpha\sigma|\mu(i)| \leq \widehat{\Sigma}_n(i,i)$$

and $i \in \widehat{S}_n$ (we ignore strict equality above as a measure 0 event), i.e. $\widetilde{S}(\theta) \subseteq \widehat{S}_n$, which concludes the proof. $\square$

## 2.8 Sparse upper bound

**Theorem 2.** *For any* $\theta = (\mu_0 - \mu, \mu_0 + \mu) \in \Theta_{\lambda,s}$ *and* $X_1, ..., X_n \overset{i.i.d.}{\sim} P_\theta$ *with* $n \geq \max(68, 4s)$ *and* $s \geq 1$, *define*

$$\alpha = \sqrt{\frac{6\log(nd)}{n}} + \frac{2\log(nd)}{n},$$

$$\widehat{\tau}_n = \frac{1+\alpha}{1-\alpha} \min_{i \in [d]} \widehat{\Sigma}_n(i,i),$$

*and*

$$\widehat{S}_n = \{i \in [d] : \widehat{\Sigma}_n(i,i) > \widehat{\tau}_n\}.$$

*Assume that* $d \geq 2$, *and* $\alpha \leq \frac{1}{4}$. *Let*

$$\widehat{F}_n(x) = \begin{cases} 1 & \text{if } x^T_{\widehat{S}_n} v_1(\widehat{\Sigma}_{\widehat{S}_n}) \geq \widehat{\mu}^T_{\widehat{S}_n} v_1(\widehat{\Sigma}_{\widehat{S}_n}) \\ 2 & \text{otherwise} \end{cases}$$

*where* $\widehat{\mu}_{\widehat{S}_n}$ *and* $\widehat{\Sigma}_{\widehat{S}_n}$ *are the empirical mean and covariance of* $X_i$ *for the dimensions in* $\widehat{S}_n$, *and* $0$ *elsewhere. Then*

$$\sup_{\theta \in \Theta_{\lambda,s}} \mathbb{E}L_\theta(\widehat{F}) \leq 603 \max\left(\frac{16\sigma^2}{\lambda^2}, 1\right) \sqrt{\frac{s\log(ns)}{n}} + 220\frac{\sigma\sqrt{s}}{\lambda}\left(\frac{\log(nd)}{n}\right)^{\frac{1}{4}}.$$

*Proof.* Define
$$S(\theta) = \{i \in [d] : \mu(i) \neq 0\}$$
and
$$\widetilde{S}(\theta) = \{i \in [d] : |\mu(i)| \geq 4\sigma\sqrt{\alpha}\},$$
Assume $\widetilde{S}(\theta) \subseteq \widehat{S}_n \subseteq S(\theta)$ (by Proposition 11, this holds with probability at least $1 - \frac{6}{n}$). If $\widetilde{S}(\theta) = \emptyset$, then we simply have $\mathbb{E}L_\theta(\widehat{F}_n) \leq \frac{1}{2}$.

Assume $\widetilde{S}(\theta) \neq \emptyset$. Let
$$\cos\widehat{\beta} = |v_1(\widehat{\Sigma}_{\widehat{S}_n})^T v_1(\Sigma)|,$$
$$\cos\widetilde{\beta} = |v_1(\Sigma_{\widehat{S}_n})^T v_1(\Sigma)|,$$
and
$$\cos\beta = |v_1(\widehat{\Sigma}_{\widehat{S}_n})^T v_1(\Sigma_{\widehat{S}_n})|$$
where $\Sigma = \sigma^2 I + \mu\mu^T$, and $\Sigma_{\widehat{S}_n}$ is the same as $\Sigma$ in $\widehat{S}_n$, and 0 elsewhere. Then
$$\sin\widehat{\beta} \leq \sin\widetilde{\beta} + \sin\beta.$$

Also
$$\sin\widetilde{\beta} = \frac{\|\mu - \mu_{\widehat{S}(\theta)}\|}{\|\mu\|}$$
$$\leq \frac{\|\mu - \mu_{\widetilde{S}(\theta)}\|}{\|\mu\|}$$
$$\leq \frac{4\sigma\sqrt{\alpha}\sqrt{|S(\theta)| - |\widetilde{S}(\theta)|}}{\|\mu\|}$$
$$\leq 8\frac{\sigma\sqrt{s\alpha}}{\lambda}.$$

Using the same argument as the proof of Theorem 1, we have that as long as the above bound is smaller than $\frac{1}{2\sqrt{5}}$,
$$\mathbb{E}L_\theta(\widehat{F}) \leq 600 \max\left(\frac{\sigma^2}{\left(\frac{\lambda}{2} - 4\sigma\sqrt{s\alpha}\right)^2}, 1\right)\sqrt{\frac{s\log(ns)}{n}} + 104\frac{\sigma\sqrt{s\alpha}}{\lambda} + \frac{3}{n}$$
$$\leq 603 \max\left(16\frac{\sigma^2}{\lambda^2}, 1\right)\sqrt{\frac{s\log(ns)}{n}} + 104\frac{\sigma\sqrt{s\alpha}}{\lambda}.$$

However, when $8\frac{\sigma\sqrt{s\alpha}}{\lambda} > \frac{1}{2\sqrt{5}}$, the above bound is bigger than $\frac{1}{2}$, which is a trivial upper bound on the clustering error, hence the bound can be stated without further conditions. Finally, since $\alpha \leq \frac{1}{4}$, we must have $\frac{\log(nd)}{n} \leq 1$, so $\alpha \leq (\sqrt{6} + 2)\sqrt{\frac{\log(nd)}{n}}$, which completes the proof.

$\square$

# 3 Lower bounds

## 3.1 Standard tools

**Lemma 2.** *Let $P_0, P_1, ..., P_M$ be probability measures satisfying*
$$\frac{1}{M}\sum_{i=1}^{M}\mathrm{KL}(P_i, P_0) \leq \alpha \log M$$
*where $0 < \alpha < 1/8$ and $M \geq 2$. Then*
$$\inf_\psi \max_{i \in [0..M]} P_i(\psi \neq i) \geq 0.07$$
*(Tsybakov (2009)).*

**Lemma 3.** *(Varshamov–Gilbert bound) Let $\Omega = \{0,1\}^m$ for $m \geq 8$. Then there exists a subset $\{\omega_0, ..., \omega_M\} \subseteq \Omega$ such that $\omega_0 = (0, ..., 0)$,*

$$\rho(\omega_i, \omega_j) \geq \frac{m}{8}, \quad \forall \, 0 \leq i < j \leq M,$$

*and*

$$M \geq 2^{m/8},$$

*where $\rho$ denotes the Hamming distance between two vectors (Tsybakov (2009)).*

**Lemma 4.** *Let $\Omega = \{\omega \in \{0,1\}^m : \|\omega\|_0 = s\}$ for integers $m > s \geq 1$. For any $\alpha, \beta \in (0,1)$ such that $s \leq \alpha \beta m$, there exists $\omega_0, ..., \omega_M \in \Omega$ such that for all $0 \leq i < j \leq M$,*

$$\rho(\omega_i, \omega_j) > 2(1-\alpha)s$$

*and*

$$\log(M+1) \geq cs \log\left(\frac{m}{s}\right)$$

*where*

$$c = \frac{\alpha}{-\log(\alpha\beta)}(-\log\beta + \beta - 1).$$

*In particular, setting $\alpha = 3/4$ and $\beta = 1/3$, we have that $\rho(\omega_i, \omega_j) > s/2$, $\log(M+1) \geq \frac{s}{5}\log\left(\frac{m}{s}\right)$, as long as $s \leq m/4$ (Massart (2007), Lemma 4.10).*

## 3.2 A reduction to hypothesis testing without a general triangle inequality

**Proposition 12.** *Let $\theta_0, ..., \theta_M \in \Theta_\lambda$ (or $\Theta_{\lambda,s}$), $M \geq 2$, $0 < \alpha < 1/8$, and $\gamma > 0$. If*

$$\max_{i \in [M]} \mathrm{KL}(P_{\theta_i}, P_{\theta_0}) \leq \frac{\alpha \log M}{n}$$

*and for all $0 \leq i \neq j \leq M$ and clusterings $\widehat{F}$,*

$$L_{\theta_i}(\widehat{F}) < \gamma \text{ implies } L_{\theta_j}(\widehat{F}) \geq \gamma,$$

*then*

$$\inf_{\widehat{F}_n} \max_{i \in [0..M]} \mathbb{E}_{\theta_i} L_{\theta_i}(\widehat{F}_n) \geq 0.07\gamma.$$

*Proof.* Using Markov's inequality,

$$\inf_{\widehat{F}_n} \max_{i \in [0..M]} \mathbb{E}_{\theta_i} L_{\theta_i}(\widehat{F}_n) \geq \gamma \inf_{\widehat{F}_n} \max_{i \in [0..M]} P_{\theta_i}^n \left( L_{\theta_i}(\widehat{F}_n) \geq \gamma \right).$$

Define $\psi^*(\widehat{F}_n) = \operatorname*{argmin}_{i \in [0..M]} L_{\theta_i}(\widehat{F}_n)$. By assumption, $L_{\theta_i}(\widehat{F}_n) < \gamma$ implies $L_{\theta_j}(\widehat{F}_n) \geq \gamma$ for any $j \neq i$, so $L_{\theta_i}(\widehat{F}_n) < \gamma$ only when $\psi^*(\widehat{F}_n) = i$. Hence,

$$P_{\theta_i}^n \left( \psi^*(\widehat{F}_n) = i \right) \geq P_{\theta_i}^n \left( L_{\theta_i}(\widehat{F}_n) < \gamma \right)$$

and

$$\inf_{\widehat{F}_n} \max_{i \in [0..M]} P_{\theta_i}^n \left( L_{\theta_i}(\widehat{F}_n) \geq \gamma \right) \geq \max_{i \in [0..M]} P_{\theta_i}^n \left( \psi^*(\widehat{F}_n) \neq i \right)$$

$$\geq \inf_{\widehat{\psi}_n} \max_{i \in [0..M]} P_{\theta_i}^n \left( \widehat{\psi}_n \neq i \right)$$

$$\geq 0.07$$

where the last step is by Lemma 2. $\qquad \square$

### 3.3 Properties of the clustering error

**Proposition 13.** *For any $\theta, \theta' \in \Theta_\lambda$, and any clustering $\widehat{F}$, if*

$$L_\theta(F_{\theta'}) + L_\theta(\widehat{F}) + \sqrt{\frac{\mathrm{KL}(P_\theta, P_{\theta'})}{2}} \leq \frac{1}{2},$$

*then*

$$L_\theta(F_{\theta'}) - L_\theta(\widehat{F}) - \sqrt{\frac{\mathrm{KL}(P_\theta, P_{\theta'})}{2}} \leq L_{\theta'}(\widehat{F}) \leq L_\theta(F_{\theta'}) + L_\theta(\widehat{F}) + \sqrt{\frac{\mathrm{KL}(P_\theta, P_{\theta'})}{2}}.$$

*Proof.* WLOG assume $F_\theta$, $F_{\theta'}$, and $\widehat{F}$ are such that, using simplified notation,

$$L_\theta(F_{\theta'}) = P_\theta(F_\theta \neq F_{\theta'})$$

and

$$L_\theta(\widehat{F}) = P_\theta(F_\theta \neq \widehat{F}).$$

Then

$$P_\theta(F_{\theta'} \neq \widehat{F}) = P_\theta\left((F_\theta = F_{\theta'}) \cap (F_\theta \neq \widehat{F}) \cup (F_\theta \neq F_{\theta'}) \cap (F_\theta = \widehat{F})\right)$$

$$= P_\theta\left((F_\theta = F_{\theta'}) \cap (F_\theta \neq \widehat{F})\right) + P_\theta\left((F_\theta \neq F_{\theta'}) \cap (F_\theta = \widehat{F})\right).$$

Since

$$0 \leq P_\theta\left((F_\theta = F_{\theta'}) \cap (F_\theta \neq \widehat{F})\right) \leq P_\theta\left(F_\theta \neq \widehat{F}\right) = L_\theta(\widehat{F}),$$

$$P_\theta\left((F_\theta \neq F_{\theta'}) \cap (F_\theta = \widehat{F})\right) \leq P_\theta\left(F_\theta \neq F_{\theta'}\right) = L_\theta(F_{\theta'}),$$

and

$$L_\theta(F_{\theta'}) - L_\theta(\widehat{F}) = P_\theta\left(F_\theta \neq F_{\theta'}\right) - P_\theta(F_\theta \neq \widehat{F}) \leq P_\theta\left((F_\theta \neq F_{\theta'}) \cap (F_\theta = \widehat{F})\right),$$

we have that

$$L_\theta(F_{\theta'}) - L_\theta(\widehat{F}) \leq P_\theta(F_{\theta'} \neq \widehat{F}) \leq L_\theta(F_{\theta'}) + L_\theta(\widehat{F})$$

and

$$L_\theta(F_{\theta'}) - L_\theta(\widehat{F}) - \mathrm{TV}(P_\theta, P_{\theta'}) \leq P_{\theta'}(F_{\theta'} \neq \widehat{F}) \leq L_\theta(F_{\theta'}) + L_\theta(\widehat{F}) + \mathrm{TV}(P_\theta, P_{\theta'}).$$

It is easy to see that if $L_\theta(F_{\theta'}) + L_\theta(\widehat{F}) + \mathrm{TV}(P_\theta, P_{\theta'}) \leq \frac{1}{2}$, then the above bound implies

$$L_\theta(F_{\theta'}) - L_\theta(\widehat{F}) - \mathrm{TV}(P_\theta, P_{\theta'}) \leq L_{\theta'}(\widehat{F}) \leq L_\theta(F_{\theta'}) + L_\theta(\widehat{F}) + \mathrm{TV}(P_\theta, P_{\theta'}).$$

The final step is to use the fact that $\mathrm{TV}(P_\theta, P_{\theta'}) \leq \sqrt{\frac{\mathrm{KL}(P_\theta, P_{\theta'})}{2}}$. $\qquad\square$

**Proposition 14.** *For some $\mu_0, \mu, \mu' \in \mathbb{R}^d$ such that $\|\mu\| = \|\mu'\|$, let*

$$\theta = \left(\mu_0 - \frac{\mu}{2}, \mu_0 + \frac{\mu}{2}\right)$$

*and*

$$\theta' = \left(\mu_0 - \frac{\mu'}{2}, \mu_0 + \frac{\mu'}{2}\right).$$

*Then*

$$2g\left(\frac{\|\mu\|}{2\sigma}\right)\sin\beta\cos\beta \leq L_\theta(F_{\theta'}) \leq \frac{1}{\pi}\tan\beta$$

*where $\cos\beta = \frac{|\mu^T \mu'|}{\|\mu\|^2}$ and $g(x) = \phi(x)(\phi(x) - x\Phi(-x))$.*

*Proof.* It is easy to see that

$$L_\theta(F_{\theta'}) = \frac{1}{2} \int_{\mathbb{R}} \frac{1}{\sigma} \phi\left(\frac{x}{\sigma}\right) \left( \Phi\left(\frac{\|\mu\|}{2\sigma} + \frac{|x|\tan\beta}{\sigma}\right) - \Phi\left(\frac{\|\mu\|}{2\sigma} - \frac{|x|\tan\beta}{\sigma}\right) \right) dx.$$

Define $\xi = \frac{\|\mu\|}{2\sigma}$. With a change of variables, we have

$$L_\theta(F_{\theta'}) = \frac{1}{2} \int_{\mathbb{R}} \phi(x) \left( \Phi\left(\xi + |x|\tan\beta\right) - \Phi\left(\xi - |x|\tan\beta\right) \right) dx$$

$$= \int_0^\infty \phi(x)(\Phi(\xi + x\tan\beta) - \Phi(\xi - x\tan\beta))dx.$$

For any $a \le b$, $\Phi(b) - \Phi(a) \le \frac{b-a}{\sqrt{2\pi}}$, so

$$L_\theta(F_{\theta'}) = \int_0^\infty \phi(x)(\Phi(\xi + x\tan\beta) - \Phi(\xi - x\tan\beta))dx$$

$$\le \int_0^\infty \phi(x)(\Phi(x\tan\beta) - \Phi(-x\tan\beta))dx$$

$$\le \tan\beta \sqrt{\frac{2}{\pi}} \int_0^\infty x\phi(x)dx$$

$$= \frac{1}{\pi}\tan\beta.$$

Also,

$$L_\theta(F_{\theta'}) = \int_0^\infty \phi(x)(\Phi(\xi + x\tan\beta) - \Phi(\xi - x\tan\beta))dx$$

$$\ge 2\tan\beta \int_0^\infty x\phi(x)\phi(\xi + x\tan\beta)dx$$

$$= 2\tan\beta \frac{1}{\sqrt{2\pi}} \int_0^\infty x\frac{1}{\sqrt{2\pi}} \exp\left\{ -\frac{x^2 + (\xi + x\tan\beta)^2}{2} \right\} dx$$

$$= 2\tan\beta \frac{1}{\sqrt{2\pi}} \exp\left\{ -\frac{\xi^2}{2}\left(1 - \frac{\tan^2\beta}{1+\tan^2\beta}\right) \right\} \int_0^\infty x\frac{1}{\sqrt{2\pi}} \exp\left\{ -\frac{\left(x + \frac{\xi\tan\beta}{1+\tan^2\beta}\right)^2}{2\left(\frac{1}{\sqrt{1+\tan^2\beta}}\right)^2} \right\} dx$$

$$\ge 2\tan\beta \frac{1}{\sqrt{2\pi}} \exp\left\{ -\frac{\xi^2}{2} \right\} \int_0^\infty x\frac{1}{\sqrt{2\pi}} \exp\left\{ -\frac{(x + \xi\sin\beta\cos\beta)^2}{2\cos^2\beta} \right\} dx$$

$$= 2\tan\beta\phi(\xi) \left[ \frac{\cos^2\beta}{\sqrt{2\pi}} \exp\left\{ -\frac{\xi^2\sin^2\beta}{2} \right\} - \xi\sin\beta\cos^2\beta\Phi(-\xi\sin\beta) \right]$$

$$= 2\sin\beta\cos\beta\phi(\xi) \left[ \phi(\xi\sin\beta) - \xi\sin\beta\Phi(-\xi\sin\beta) \right]$$

$$\ge 2\sin\beta\cos\beta\phi(\xi) \left[ \phi(\xi) - \xi\Phi(-\xi) \right].$$

$\square$

## 3.4 A KL divergence bound of the necessary order

**Proposition 15.** *For some $\mu_0, \mu, \mu' \in \mathbb{R}^d$ such that $\|\mu\| = \|\mu'\|$, let*

$$\theta = \left(\mu_0 - \frac{\mu}{2}, \mu_0 + \frac{\mu}{2}\right)$$

*and*

$$\theta' = \left(\mu_0 - \frac{\mu'}{2}, \mu_0 + \frac{\mu'}{2}\right).$$

*Then*

$$\mathrm{KL}(P_\theta, P_{\theta'}) \le \xi^4(1 - \cos\beta)$$

*where $\xi = \frac{\|\mu\|}{2\sigma}$ and $\cos\beta = \frac{|\mu^T \mu'|}{\|\mu\|\|\mu'\|}$.*

*Proof.* Since the KL divergence is invariant to affine transformations, it is easy to see that

$$\mathrm{KL}(P_\theta, P_{\theta'}) = \int_\mathbb{R} \int_\mathbb{R} p_1(x, y) \log \frac{p_1(x, y)}{p_2(x, y)} dx dy$$

where

$$p_1(x, y) = \frac{1}{2}\phi(x + \xi_x)\phi(y + \xi_y) + \frac{1}{2}\phi(x - \xi_x)\phi(y - \xi_y),$$

$$p_2(x, y) = \frac{1}{2}\phi(x + \xi_x)\phi(y - \xi_y) + \frac{1}{2}\phi(x - \xi_x)\phi(y + \xi_y),$$

$$\xi_x = \xi \cos\frac{\beta}{2}, \quad \xi_y = \xi \sin\frac{\beta}{2}.$$

Since

$$\begin{aligned}
\frac{p_1(x, y)}{p_2(x, y)} &= \frac{\phi(x + \xi_x)\phi(y + \xi_y) + \phi(x - \xi_x)\phi(y - \xi_y)}{\phi(x + \xi_x)\phi(y - \xi_y) + \phi(x - \xi_x)\phi(y + \xi_y)} \\
&= \frac{\exp(-x\xi_x - y\xi_y) + \exp(x\xi_x + y\xi_y)}{\exp(-x\xi_x + y\xi_y) + \exp(x\xi_x - y\xi_y)}
\end{aligned}$$

we have

$$\log \frac{p_1(x, y)}{p_2(x, y)} = \log \frac{\cosh(x\xi_x + y\xi_y)}{\cosh(x\xi_x - y\xi_y)}.$$

Furthermore,

$$\begin{aligned}
&\int_\mathbb{R} \int_\mathbb{R} \frac{1}{2}\phi(x + \xi_x)\phi(y + \xi_y) \log \frac{\cosh(x\xi_x + y\xi_y)}{\cosh(x\xi_x - y\xi_y)} dx dy \\
&= \int_\mathbb{R} \int_\mathbb{R} \frac{1}{2}\phi(-x + \xi_x)\phi(-y + \xi_y) \log \frac{\cosh(-x\xi_x - y\xi_y)}{\cosh(-x\xi_x + y\xi_y)} dx dy \\
&= \int_\mathbb{R} \int_\mathbb{R} \frac{1}{2}\phi(x - \xi_x)\phi(y - \xi_y) \log \frac{\cosh(x\xi_x + y\xi_y)}{\cosh(x\xi_x - y\xi_y)} dx dy
\end{aligned}$$

so

$$\begin{aligned}
\mathrm{KL}(P_\theta, P_{\theta'}) &= \int_\mathbb{R} \int_\mathbb{R} \phi(x - \xi_x)\phi(y - \xi_y) \log \frac{\cosh(x\xi_x + y\xi_y)}{\cosh(x\xi_x - y\xi_y)} dx dy \\
&= \int_\mathbb{R} \int_\mathbb{R} \phi(x)\phi(y) \log \frac{\cosh(x\xi_x + \xi_x^2 + y\xi_y + \xi_y^2)}{\cosh(x\xi_x + \xi_x^2 - y\xi_y - \xi_y^2)} dx dy.
\end{aligned}$$

But for any $x$

$$-\int_{\mathbb{R}} \phi(x)\phi(y) \log \cosh(x\xi_x + \xi_x^2 - y\xi_y - \xi_y^2) dy$$

$$= -\int_{\mathbb{R}} \phi(x)\phi(-y) \log \cosh(x\xi_x + \xi_x^2 + y\xi_y - \xi_y^2) dy$$

$$= -\int_{\mathbb{R}} \phi(x)\phi(y) \log \cosh(x\xi_x + \xi_x^2 + y\xi_y - \xi_y^2) dy,$$

thus,

$$\mathrm{KL}(P_\theta, P_{\theta'}) = \int_{\mathbb{R}} \int_{\mathbb{R}} \phi(x)\phi(y) \log \frac{\cosh(x\xi_x + \xi_x^2 + y\xi_y + \xi_y^2)}{\cosh(x\xi_x + \xi_x^2 + y\xi_y - \xi_y^2)} dx dy$$

$$= \int_{\mathbb{R}} \phi(z) \log \frac{\cosh(z\sqrt{\xi_x^2 + \xi_y^2} + \xi_x^2 + \xi_y^2)}{\cosh(z\sqrt{\xi_x^2 + \xi_y^2} + \xi_x^2 - \xi_y^2)} dz$$

$$= \int_{\mathbb{R}} \phi(z) \log \frac{\cosh(\xi z + \xi_x^2 + \xi_y^2)}{\cosh(\xi z + \xi_x^2 - \xi_y^2)} dz$$

since $\xi_x^2 + \xi_y^2 = \xi^2$. By the mean value theorem and the fact that $\tanh$ is monotonically increasing,

$$\log \frac{\cosh(\xi z + \xi_x^2 + \xi_y^2)}{\cosh(\xi z + \xi_x^2 - \xi_y^2)} \le 2\xi_y^2 \tanh(\xi z + \xi_x^2 + \xi_y^2)$$

$$= 2\xi_y^2 \tanh(\xi z + \xi^2)$$

for all $z$. Since $\tanh$ is an odd function,

$$\mathrm{KL}(P_\theta, P_{\theta'}) \le 2\xi_y^2 \int_{\mathbb{R}} \phi(z) \tanh(\xi z + \xi^2) dz$$

$$= 2\xi_y^2 \int_{\mathbb{R}} \phi(z)(\tanh(\xi z + \xi^2) - \tanh(\xi z)) dz.$$

Using the mean value theorem again,

$$\tanh(\xi z + \xi^2) - \tanh(\xi z) \le \xi^2 \max_{x \in [\xi z, \xi z + \xi^2]} (1 - \tanh^2(x))$$

$$\le \xi^2$$

for all $z$, so

$$\mathrm{KL}(P_\theta, P_{\theta'}) \le 2\xi^2 \xi_y^2$$

$$= 2\xi^4 \sin^2 \frac{\beta}{2}$$

$$= \xi^4 (1 - \cos \beta).$$

$\square$

## 3.5    Non-sparse lower bound

**Theorem 3.** *Assume that $d \ge 9$ and $\frac{\lambda}{\sigma} \le 0.2$. Then*

$$\inf_{\widehat{F}_n} \sup_{\theta \in \Theta_\lambda} \mathbb{E}_\theta L_\theta(\widehat{F}_n) \ge \frac{1}{500} \min \left\{ \frac{\sqrt{\log 2}}{3} \frac{\sigma^2}{\lambda^2} \sqrt{\frac{d-1}{n}}, \frac{1}{4} \right\}.$$

*Proof.* Let $\xi = \frac{\lambda}{2\sigma}$, and define

$$\epsilon = \min\left\{\frac{\sqrt{\log 2}}{3}\frac{\sigma^2}{\lambda}\frac{1}{\sqrt{n}}, \frac{\lambda}{4\sqrt{d-1}}\right\}.$$

Define $\lambda_0^2 = \lambda^2 - (d-1)\epsilon^2$. Let $\Omega = \{0,1\}^{d-1}$. For $\omega = (\omega(1), ..., \omega(d-1)) \in \Omega$, let $\mu_\omega = \lambda_0 e_d + \sum_{i=1}^{d-1}(2\omega(i)-1)\epsilon e_i$ (where $\{e_i\}_{i=1}^d$ is the standard basis for $\mathbb{R}^d$). Let $\theta_\omega = \left(-\frac{\mu_\omega}{2}, \frac{\mu_\omega}{2}\right) \in \Theta_\lambda$.

By Proposition 15, for any $\omega, \nu \in \Omega$,

$$\mathrm{KL}(P_{\theta_\omega}, P_{\theta_\nu}) \le \xi^4(1 - \cos\beta_{\omega,\nu})$$

where

$$\cos\beta_{\omega,\nu} = \frac{|\mu_\omega^T\mu_\nu|}{\lambda^2} = 1 - \frac{2\rho(\omega,\nu)\epsilon^2}{\lambda^2}$$

and $\rho$ is the Hamming distance, so

$$\mathrm{KL}(P_{\theta_\omega}, P_{\theta_\nu}) \le \xi^4\frac{2\rho(\omega,\nu)\epsilon^2}{\lambda^2}$$

$$\le \xi^4\frac{2(d-1)\epsilon^2}{\lambda^2}.$$

By Proposition 14, since $\cos\beta_{\omega,\nu} \ge \frac{1}{2}$,

$$L_{\theta_\omega}(F_{\theta_\nu}) \le \frac{1}{\pi}\tan\beta_{\omega,\nu}$$

$$\le \frac{1}{\pi}\frac{\sqrt{1 + \cos\beta_{\omega,\nu}}}{\cos\beta_{\omega,\nu}}\sqrt{1 - \cos\beta_{\omega,\nu}}$$

$$\le \frac{4}{\pi}\frac{\sqrt{d-1}\epsilon}{\lambda}$$

and

$$L_{\theta_\omega}(F_{\theta_\nu}) \ge 2g(\xi)\sin\beta_{\omega,\nu}\cos\beta_{\omega,\nu}$$

$$\ge g(\xi)\sin\beta_{\omega,\nu}$$

$$\ge \sqrt{2}g(\xi)\frac{\sqrt{\rho(\omega,\nu)}\epsilon}{\lambda}$$

where $g(x) = \phi(x)(\phi(x) - x\Phi(-x))$. By Lemma 3, there exist $\omega_0, ..., \omega_M \in \Omega$ such that $M \ge 2^{(d-1)/8}$ and

$$\rho(\omega_i, \omega_j) \ge \frac{d-1}{8}, \quad \forall\, 0 \le i < j \le M.$$

For simplicity of notation, let $\theta_i = \theta_{\omega_i}$ for all $i \in [0..M]$. Then, for $i \ne j \in [0..M]$,

$$\mathrm{KL}(P_{\theta_i}, P_{\theta_j}) \le \xi^4\frac{2(d-1)\epsilon^2}{\lambda^2},$$

and

$$L_{\theta_i}(F_{\theta_j}) \le \frac{4}{\pi}\frac{\sqrt{d-1}\epsilon}{\lambda}$$

and

$$L_{\theta_i}(F_{\theta_j}) \ge \frac{1}{2}g(\xi)\frac{\sqrt{d-1}\epsilon}{\lambda}.$$

Define

$$\gamma = \frac{1}{4}(g(\xi) - 2\xi^2)\frac{\sqrt{d-1}\epsilon}{\lambda}.$$

Then for any $i \neq j \in [0..M]$, and any $\widehat{F}$ such that $L_{\theta_i}(\widehat{F}) < \gamma$,

$$L_{\theta_i}(F_{\theta_j}) + L_{\theta_i}(\widehat{F}) + \sqrt{\frac{\mathrm{KL}(P_{\theta_i}, P_{\theta_j})}{2}} < \left(\frac{4}{\pi} + \frac{1}{4}(g(\xi) - 2\xi^2) + \xi^2\right)\frac{\sqrt{d-1}\epsilon}{\lambda} \leq \frac{1}{2}$$

because, for $\xi \leq 0.1$, by definition of $\epsilon$,

$$\left(\frac{4}{\pi} + \frac{1}{4}(g(\xi) - 2\xi^2) + \xi^2\right)\frac{\sqrt{d-1}\epsilon}{\lambda} \leq 2\frac{\sqrt{d-1}\epsilon}{\lambda} \leq \frac{1}{2}.$$

So, by Proposition 13,

$$L_{\theta_j}(\widehat{F}) \geq L_{\theta_i}(F_{\theta_j}) - L_{\theta_i}(\widehat{F}) - \sqrt{\frac{\mathrm{KL}(P_{\theta_i}, P_{\theta_j})}{2}} \geq \gamma.$$

Also,

$$\max_{i \in [M]} \mathrm{KL}(P_{\theta_i}, P_{\theta_0}) \leq (d-1)\xi^4\frac{2\epsilon^2}{\lambda^2}$$

$$\leq \frac{\log M}{9n}$$

because, by definition of $\epsilon$,

$$\xi^4\frac{2\epsilon^2}{\lambda^2} \leq \frac{\log 2}{72n}.$$

So by Proposition 12 and the fact that $\xi \leq 0.1$,

$$\inf_{\widehat{F}_n} \max_{i \in [0..M]} \mathbb{E}_{\theta_i} L_{\theta_i}(\widehat{F}_n) \geq 0.07\gamma$$

$$= 0.07\frac{1}{4}(g(\xi) - 2\xi^2)\frac{\sqrt{d-1}\epsilon}{\lambda}$$

$$\geq \frac{1}{500}\min\left\{\frac{\sqrt{\log 2}}{3}\frac{\sigma^2}{\lambda^2}\sqrt{\frac{d-1}{n}}, \frac{1}{4}\right\}$$

and to complete the proof we use the fact that

$$\inf_{\widehat{F}_n} \sup_{\theta \in \Theta_\lambda} \mathbb{E}_\theta L_\theta(\widehat{F}_n) \geq \inf_{\widehat{F}_n} \max_{i \in [0..M]} \mathbb{E}_{\theta_i} L_{\theta_i}(\widehat{F}_n).$$

$\square$

## 3.6 Sparse lower bound

**Theorem 4.** *Assume that $\frac{\lambda}{\sigma} \leq 0.2$, $d \geq 17$, and*

$$5 \leq s \leq \frac{d-1}{4} + 1.$$

*Then*

$$\inf_{\widehat{F}_n} \sup_{\theta \in \Theta_{\lambda,s}} \mathbb{E}_\theta L_\theta(\widehat{F}_n) \geq \frac{1}{600}\min\left\{\sqrt{\frac{8}{45}}\frac{\sigma^2}{\lambda^2}\sqrt{\frac{s-1}{n}\log\left(\frac{d-1}{s-1}\right)}, \frac{1}{2}\right\}.$$

*Proof.* For simplicity, we state this proof for $\Theta_{\lambda,s+1}$, assuming $4 \leq s \leq \frac{d-1}{4}$. Let $\xi = \frac{\lambda}{2\sigma}$, and define

$$\epsilon = \min\left\{\sqrt{\frac{8}{45}}\frac{\sigma^2}{\lambda}\sqrt{\frac{1}{n}\log\left(\frac{d-1}{s}\right)}, \frac{1}{2}\frac{\lambda}{\sqrt{s}}\right\}.$$

Define $\lambda_0^2 = \lambda^2 - s\epsilon^2$. Let $\Omega = \{\omega \in \{0,1\}^{d-1} : \|\omega\|_0 = s\}$. For $\omega = (\omega(1), ..., \omega(d-1)) \in \Omega$, let $\mu_\omega = \lambda_0 e_d + \sum_{i=1}^{d-1}\omega(i)\epsilon e_i$ (where $\{e_i\}_{i=1}^d$ is the standard basis for $\mathbb{R}^d$). Let $\theta_\omega = \left(-\frac{\mu_\omega}{2}, \frac{\mu_\omega}{2}\right) \in \Theta_{\lambda,s}$.

By Proposition 15,

$$\mathrm{KL}(P_{\theta_\omega}, P_{\theta_\nu}) \leq \xi^4(1 - \cos\beta_{\omega,\nu})$$

where

$$\cos\beta_{\omega,\nu} = \frac{|\mu_\omega^T \mu_\nu|}{\lambda^2} = 1 - \frac{\rho(\omega,\nu)\epsilon^2}{2\lambda^2}$$

and $\rho$ is the Hamming distance, so

$$\mathrm{KL}(P_{\theta_\omega}, P_{\theta_\nu}) \leq \xi^4 \frac{\rho(\omega,\nu)\epsilon^2}{2\lambda^2}$$
$$\leq \xi^4 \frac{s\epsilon^2}{\lambda^2}.$$

By Proposition 14, since $\cos\beta_{\omega,\nu} \geq \frac{1}{2}$,

$$L_{\theta_\omega}(F_{\theta_\nu}) \leq \frac{1}{\pi}\tan\beta_{\omega,\nu}$$
$$\leq \frac{2}{\pi}\sin\beta_{\omega,\nu}$$
$$\leq \frac{2\sqrt{2}}{\pi}\frac{\sqrt{s}\epsilon}{\lambda}$$

and

$$L_{\theta_\omega}(F_{\theta_\nu}) \geq 2g(\xi)\sin\beta_{\omega,\nu}\cos\beta_{\omega,\nu}$$
$$\geq g(\xi)\sin\beta_{\omega,\nu}$$
$$\geq \frac{g(\xi)}{\sqrt{2}}\frac{\sqrt{\rho(\omega,\nu)}\epsilon}{\lambda}$$

where $g(x) = \phi(x)(\phi(x) - x\Phi(-x))$. By Lemma 4, there exist $\omega_0, ..., \omega_M \in \Omega$ such that $\log(M + 1) \geq \frac{s}{5}\log\left(\frac{d-1}{s}\right)$ and

$$\rho(\omega_i, \omega_j) \geq \frac{s}{2}, \quad \forall\, 0 \leq i < j \leq M.$$

For simplicity of notation, let $\theta_i = \theta_{\omega_i}$ for all $i \in [0..M]$. Then, for $i \neq j \in [0..M]$,

$$\mathrm{KL}(P_{\theta_i}, P_{\theta_j}) \leq \xi^4 \frac{s\epsilon^2}{\lambda^2},$$

and

$$L_{\theta_i}(F_{\theta_j}) \leq \frac{2\sqrt{2}}{\pi}\frac{\sqrt{s}\epsilon}{\lambda}$$

and

$$L_{\theta_i}(F_{\theta_j}) \geq \frac{g(\xi)}{2}\frac{\sqrt{s}\epsilon}{\lambda}.$$

Define

$$\gamma = \frac{1}{4}(g(\xi) - \sqrt{2}\xi^2)\frac{\sqrt{s}\epsilon}{\lambda}.$$

Then for any $i \neq j \in [0..M]$, and any $\widehat{F}$ such that $L_{\theta_i}(\widehat{F}) < \gamma$,

$$L_{\theta_i}(F_{\theta_j}) + L_{\theta_i}(\widehat{F}) + \sqrt{\frac{\mathrm{KL}(P_{\theta_i}, P_{\theta_j})}{2}} < \left(\frac{2\sqrt{2}}{\pi} + \frac{1}{4}(g(\xi) - \sqrt{2}\xi^2) + \frac{\xi^2}{\sqrt{2}}\right)\frac{\sqrt{s}\epsilon}{\lambda} \leq \frac{1}{2}$$

because, for $\xi \leq 0.1$, by definition of $\epsilon$,

$$\left(\frac{2\sqrt{2}}{\pi} + \frac{1}{4}(g(\xi) - \sqrt{2}\xi^2) + \frac{\xi^2}{\sqrt{2}}\right)\frac{\sqrt{s}\epsilon}{\lambda} \leq \frac{\sqrt{s}\epsilon}{\lambda} \leq \frac{1}{2}.$$

So, by Proposition 13,

$$L_{\theta_j}(\widehat{F}) \geq L_{\theta_i}(F_{\theta_j}) - L_{\theta_i}(\widehat{F}) - \sqrt{\frac{\mathrm{KL}(P_{\theta_i}, P_{\theta_j})}{2}} \geq \gamma.$$

Also,

$$\max_{i \in [M]} \mathrm{KL}(P_{\theta_i}, P_{\theta_0}) \leq \xi^4 \frac{s\epsilon^2}{\lambda^2}$$

$$\leq \frac{1}{18n} \log \left( \frac{d-1}{s} \right)^{\frac{s}{5}}$$

$$\leq \frac{1}{9n} \log \left( \left( \frac{d-1}{s} \right)^{\frac{s}{5}} - 1 \right)$$

$$\leq \frac{\log M}{9n}$$

because, by definition of $\epsilon$,

$$\xi^4 \frac{s\epsilon^2}{\lambda^2} \leq \frac{s}{90n} \log \left( \frac{d-1}{s} \right).$$

So by Proposition 12 and the fact that $\xi \leq 0.1$,

$$\inf_{\widehat{F}_n} \max_{i \in [0..M]} \mathbb{E}_{\theta_i} L_{\theta_i}(\widehat{F}_n) \geq 0.07\gamma$$

$$\geq 0.07 \frac{0.1}{4} \frac{\sqrt{s}\epsilon}{\lambda}$$

$$\geq \frac{1}{600} \min \left\{ \sqrt{\frac{8}{45}} \frac{\sigma^2}{\lambda^2} \sqrt{\frac{s}{n} \log \left( \frac{d-1}{s} \right)}, \frac{1}{2} \right\}$$

and to complete the proof we use the fact that

$$\inf_{\widehat{F}_n} \sup_{\theta \in \Theta_{\lambda,s}} \mathbb{E}_\theta L_\theta(\widehat{F}_n) \geq \inf_{\widehat{F}_n} \max_{i \in [0..M]} \mathbb{E}_{\theta_i} L_{\theta_i}(\widehat{F}_n).$$

$\square$