[Reviews · NeurIPS 2013]

Submitted by Assigned_Reviewer_3

The paper studies the problem of identifying Gaussians in a mixture in high dimensions when the separation between the Gaussians is small. The assumption is that the Gaussians are separated along few dimensions and hence by identifying these dimensions, that is, feature selection, the curse of dimensionality can be bitten and the Gaussians can be found.

Clustering in high dimension is an open problem that well deserve a study. The theoretical approach taken by the authors is good step in the path towards better understanding the problem. However, I have two reservations about this work. In Line 36, the authors say “there appears to be no theoretical results justifying the advantage of variable selection in high dimension setting”. As far as I understand, K. Chaudhuri and S. Rao, ("Learning Mixtures of Product Distributions using Correlations and Independence", In Proc. of Conf. on Learning Theory, 2008) tried to address similar problem of learning mixture models in high dimension under the assumption that the separation between the components is in a small, axis aligned space. While the results there are different then the results here, I was missing a comparison between the results.

In their response to the original review, the authors have addressed the differences between the current work and the work of Chaudhuri and Rao. I am confident that the work here contains novel results that should be published. However, I think that more careful thought needs to be given to presenting the novelty and contrasting it with the state of the art. The main message of the authors in their response is “these methods [the methods of Chaudhuri and Rao] do not explicitly provide variable selection guarantees, i.e. the methods are not guaranteed to correctly identify the relevant features.” However, in the paper, the authors present as the main result “In this paper, we provide precise information theoretic bounds on the clustering accuracy and sample complexity of learning a mixture of two isotropic Gaussians”. Therefore, the main goal is the accuracy of the separation as opposed to identification of relevant features. Therefore, I find the main contribution of this paper to be vague. Since I believe that this paper contains significant results, I think that the authors should reconsider the presentation to make a clear statement about the objectives and the contribution.

I also had some problems following the proof of Theorem 2 but the authors clarified it in their response.

Comments:

1. Theorem 4: The term \frac{d-1}{4}+1 could be simplified to \frac{d+3}{4}
2. Lemma 1: I believe you could improve the bound on M to $2^{7m/16}$ by a simple covering argument: $\Omega$ is of size $2^m$. Every $\omega_i$ “covers” a ball of size $m/8$. A ball of size $m/8$ contains $\sum_{i\leq m/8} \choose{m}{i}$ points from $\Omega$ which can be bounded (Sauer’s Lemma) by (8e)^{m/8} which is bounded by 2^{9m/16}. Therefore, if we choose $M=2^{7m/16}$ we have that M balls of radius $m/8$ do not cover $\Omega$ and hence there is a $m/8$ packing (which is what you are looking for) of size $M$. (This is not a major issue and you may decide to ignore it in favor of the simplicity of the statement).

3. Lemma 2: The bound on M here is stated with logarithms while the bound in Lemma 1 does not use logarithms which makes it harder to follow.

4. Proposition 3: what are $\phi$ and $\Phi$?
5. Proof of Theorem 2, line 275: What is $\theta_\nu$, where is it introduced?
6. Proof of Theorem 2, line 279: It is not clear why on the r.h.s. there is $\frac{\sqrt{d-1}\espilom}{\lambda}$. Using the term for the cosine of \beta and since $sin(\beta) = \sqrt{1-cos^2(\beta)$ I get that $sin(\beta) = \frac{2\rho\epsilon^2}{\lambda^2} \geq \frac{2(d-1)\epsilon^2}{\lambda^2}$ and this $tan{\beta} \leq \frac{4(d-1)\epsilon^2}{\lambda^2}$

8. Line 466: you use full names for all references but Tsybakov’s paper.

Summary: The paper studies lower and upper bounds for Gaussian mixture separation in high dimension under sparsity assumptions. The results are interesting but import reference is missing which leaves it to the reader to understand the novelty of the result.

Submitted by Assigned_Reviewer_5

The paper provides minimax lower bounds for the mixture of mean-separated isotropic gaussians in standard and high dimensional setting. To handle the high-dimensional case the authors propose to enforce 'sparse' mean separation - where sparsity is enforced on the difference of the two mean vectors. Furthermore simple spectral methods for model-based clustering along with sample complexity results are provided.

The proof for the non-sparse upper bound is straight forward - 118-120 is the main thing of interest which is out-sourced to existing results - I guess it would directly extend to any isotropic mixtures too. I would be surprised if no minimax lower bounds existed in the multivariate gaussian case in the standard setting - perhaps some relevant results exist in the statistics literature ?

For the sparse setting - I am inclined to agree with the authors that the more natural method of projecting onto the first sparse principal component would lead to closing the gap. I wonder why the authors did not analyze this directly method as a start ?

The proofs are mostly clear and uses standard tools/tricks. The assumption that a particular feature is irrelevant if the corresponding mean components are similar seems artificial. Can the authors provide some motivation for this ?

minor: I feel the authors could really remove the redundant sections 4 and 5 from the main paper.
Summary: The paper discusses minimax theory for the mixture of mean-separated isotropic gaussians in standard and high dimensional setting.

Submitted by Assigned_Reviewer_6

The authors study the problem of clustering from a
statistical minimax point of view. The goal is to
define a rule that assigns each example to one of
two classes given, based on a training sample drawn
from a mixture of two Gaussian distributions with
different means. The mixture weights are assumed
known and equal to 1/2, the covariance matrix of
the Gaussians is assumed proportional to the identity
matrix. The case of high-dimensionality is investigated
and matching (up to a log factor) upper and lower
bounds on the rate of separation between two means
are established.

The second contribution of the paper concerns the
minimax rates of clustering under the sparsity
assumption. Here, the rates of clustering are better
than those of previous setting provided that the
square of the sparsity is smaller than the dimension.
There is a gap between the lower bound and the upper
bound of separation: they differ by a factor s^(1/4).
Nevertheless, I find this result very interesting and
the proof of it is far from being trivial.


I have a few recommendations for improving the presentation.
1. It should be mentioned in the introduction that
the weights of the mixtures are assumed fixed and
equal to 1/2.
2. It is remarkable that the procedures considered in
sections 3.1 and 3.2 do not depend on sigma. I think that
this should be stressed in these sections.
3. Since this is a first step in a minimax study of the
clustering, I feel that the condition of known weights is
acceptable. But as far as I understand putting these
weights equal to 1/2 is not important. I strongly recommend
to add this remark along with the modification of the
procedure necessary for covering this case. (I guess that
the knowledge of sigma becomes important in this case.)

Typo: It seems that something is missing in the sentence
"The latter ..." on line 097.
Summary: The paper is very clearly written. Most results are sharp
and improves the state of the art in the statistical analysis
of clustering methods.
Author Feedback

Author rebuttal: We thank the reviewers for their comments. Below are our detailed replies.

Assigned_Reviewer_3

Thanks for pointing us to Chaudhuri and Rao COLT 2008 paper, we will add a reference to it. This paper, as well as Brubaker and Vempala 2008 (which we cite), demonstrate that the required mean separation depends on the variance along the direction/subspace separating the component means. However, these methods do not explicitly provide variable selection guarantees, i.e. the methods are not guaranteed to correctly identify the relevant features. Moreover, these papers are concerned with computational efficiency and do not give precise, statistical minimax upper and lower bounds. Specifically, the sample complexity bounds are suboptimal (both papers state that the number of samples is polynomial, however the exact dependence behaves like poly(d) instead of s log(d) or s^2 log(d) as in our results, where s is the number of relevant features). These papers also do not capture how the mean separation can shrink with increasing number of samples since they consider a different error metric – the probability of misclustering a point by their method, as opposed to our metric which considers the probability of misclutering a point by our method relative to the probability of misclustering by an oracle that knows the correct distribution. We mention all these differences with existing work in the related work section on page 2.

We agree that the form of the bound we use favors simplicity over tightness in the constants. We hope it's OK to leave the bound as is, but we will add a remark that it can be tightened.

phi and Phi refer to the univariate standard normal density and distribution functions, as defined in section 2. We will move the definition to section 4 in the revision for clarity.

$\theta_\nu$ is defined for $\nu \in \Omega$. We will clarify this.

Proof of Theorem 2, lines 279 and 281 - Essentially the bound is a consequence of the fact that cos(x)\approx 1-x^2 (up to constants), while sin(x)\approx tan(x) \approx x for small x. Stated differently, tan(x)=sqrt(1-cos(x)) * sqrt(1+cos(x)) / cos(x) = const * sqrt(1- cos(x)). We will expand the derivation in the revision to clarify.

Thanks for the other suggested improvements - we will address these in the final version.


Assigned_Reviewer_5

Learning Gaussian mixtures is a parametric problem, and hence from a statistical point of view standard parametric results for the error rates apply. However, existing results in statistics literature assume the mean separation is a constant and do not track dependence on it. For our purposes it was important to characterize the exact dependence on the mean separation. Also, these papers mostly analyze the maximum likelihood estimator that is computationally infeasible in high dimensions, and we wanted to present an efficient estimator.

For the same reason of computational feasibility, we did not analyze the combinatorial sparse SVD estimator for the sparse setting. To the best of our knowledge all current attempts at developing computationally efficient (i.e. polynomial time) algorithms for sparse SVD do not guarantee a better rate than that of the simple scheme we use in the paper. Nevertheless, we agree that demonstrating that the statistical gap can be closed by employing existing computationally inefficient methods is interesting and we are currently working on it, but time and space constraints did not allow us to address this question in our submission.

Assuming a mixture of spherical Gaussians, the relative likelihood that a given point was drawn from any particular component is constant with respect to any dimension where the means of each component are identical. In other words, the "optimal" clustering (in terms of the clustering error we use) does not depend on such directions. The underlying intuition is similar to the notion of irrelevance used in Witten and Tibshirani (2010) and Sun, Wang, and Fang (2012) for the purposes of feature selection.

While detailed proofs are provided in the appendix, we believe that the proof sketches in Sections 4 and 5 are important to provide the reader with an overview of the tools used and keep the paper self-contained.


Assigned_Reviewer_6

As the reviewer points out, the extension to unequal weights should be straight-forward and it will indeed be interesting to see if that case requires knowing the noise variance sigma. We will address this in a revised appendix.